# Organic flash memory on various flexible substrates for foldable and disposable electronics

Seungwon Lee[1], Hyejeong Seong[2], Sung Gap Im[2], Hanul Moon[1] & Seunghyup Yoo[1]

With the emergence of wearable or disposable electronics, there grows a demand for a flash memory realizable on various flexible substrates. Nevertheless, it has been challenging to develop a flash memory that simultaneously exhibits a significant level of flexibility and performance. This is mainly due to the scarcity of flexible dielectric materials with insulating properties sufficient for a flash memory, which involves dual dielectric layers, respectively, responsible for tunneling and blocking of charges. Here we report ultra-flexible organic flash memories based on polymer dielectrics prepared by initiated chemical vapor deposition. Using their near-ideal dielectric characteristics, we demonstrate flash memories bendable down to a radius of 300 μm that exhibits a relatively long-projected retention with a programming voltage on par with the present industrial standards. The proposed memory technology is then applied to non-conventional substrates, such as papers, to demonstrate its feasibility in a wide range of applications.

[1] School of Electrical Engineering, Korea Advanced Institute of Science and Technology (KAIST), Daejeon 34141, Republic of Korea. [2] Department of Chemical & Biomolecular Engineering, KAIST, Daejeon 34141, Republic of Korea. Correspondence and requests for materials should be addressed to H.M. (email: humoon@kaist.ac.kr) or to S.Y. (email: syoo@ee.kaist.ac.kr)

Flash memory is a non-volatile, transistor-based data-storage device that has become essential in most electronic systems found in modern daily life. With straightforward operation mechanisms and easy integration into NAND or NOR array architectures, flash memories have been established as by far the most successful and dominant non-volatile memory technology[1, 2]. Above all, their direct electrical programming requiring no mechanical parts greatly reduces the weight of an overall system as well as the power consumption, making it hugely popular in portable electronics spanning from thumb drives and memory cards to solid-state drives in smart phones and laptops[1, 2]. Emerging areas, such as wearable, body-attachable, or disposable electronics, will also greatly benefit from such characteristics of flash memories as well as from their well-established circuit infrastructure, provided that they can be realized on various non-rigid substrates and, at the same time, can exhibit a memory performance that is sufficient for real-world applications[3–12]. In these respects, many studies have been devoted to the development of flash memories based on organic or other emerging electronic materials that can potentially be compatible with flexible substrates[13–22].

Despite promising demonstrations in the early stages[13–36], however, the overall progress in this field has been far slower than that of thin-film transistors (TFTs) or other devices based on

flexible materials[37–46]. Major challenges lie in the core structure of flash memories, which involves dielectric bilayers sandwiching a charge storage layer such as a floating gate (FG) electrode (Fig. 1a). In addition to playing a regular role as a gate insulator, these two dielectric layers have to provide a reliable and controllable means by serving, respectively, (i) as a tunneling dielectric layer (TDL) through which to inject charges into the FG for programming or to remove charges therefrom for erasing via control of a control gate (CG) bias; and (ii) as a blocking dielectric layer (BDL) to prevent charge leakage to confine the stored charges within the FG for memory retention and to ensure that charge flow occurs only through the TDL during programming/erasing (Fig. 1b)[1, 2]. For reasonably low-programming/erasing voltages, these dielectric layers should not be made too thick; this unfortunately contradicts the conditions for long memory retention in many cases. A best-case scenario would thus be to use dielectric layers that are thin enough to enable low-voltage programming/erasing and yet can provide leak-free insulating properties even at such small thickness. Finding such an ideal dielectric layer has been very difficult for polymeric layers, which are the most prominent candidate material for flexible electronics[47–50]. The solution processing used for most polymeric dielectric layers also makes it difficult to use them in flash memories due to the complexity involved in the formation

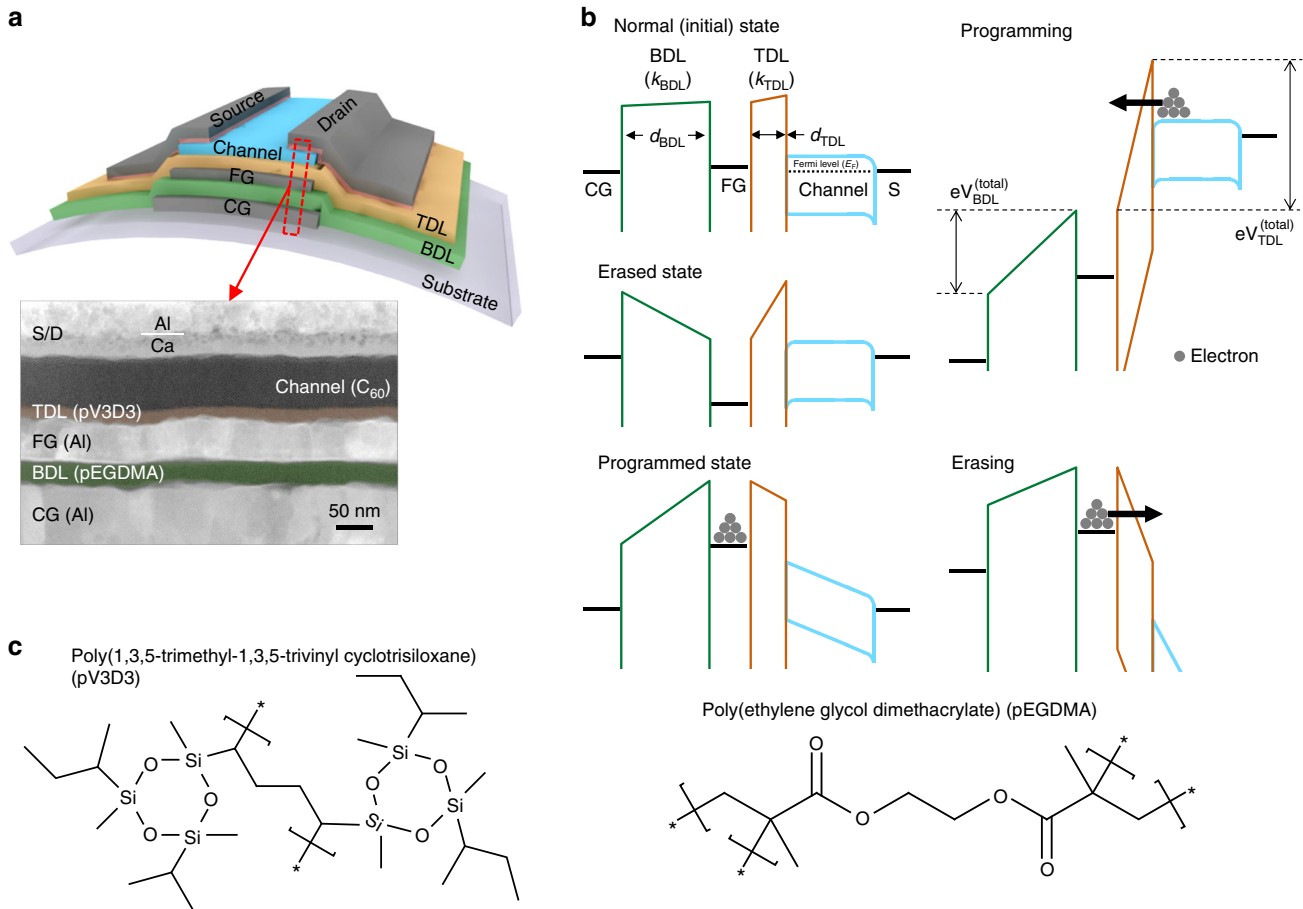

**Fig. 1** Structure of thin-film transistor-based organic flash memories and their operation mechanism. **a** The schematic device structure of the proposed flash memory and a false-color cross-sectional TEM image of the fabricated flash memory device. TDL, BDL, CG, FG, and S/D refer to tunneling dielectric layer, blocking dielectric layer, control gate, floating gate, and source/drain, respectively. **b** Energy band diagrams of the flash memory device under study for various operation regimes. The total voltages across BDL (=$V_{BDL}^{(total)}$) and TDL (=$V_{TDL}^{(total)}$) are determined by the various factors such as built-in voltages, applied $V_{CG}$, and coupling ratio, $V_{th}$ shift due to stored or depleted charges, etc., as described in details in Supplementary Table 1. **c** Chemical structures of the poly(1,3,5-trimethyl-1,3,5-trivinyl cyclotrisiloxane) (pV3D3) and the poly(ethylene glycol dimethacrylate) (pEGDMA) polymers used as TDL and BDL, respectively

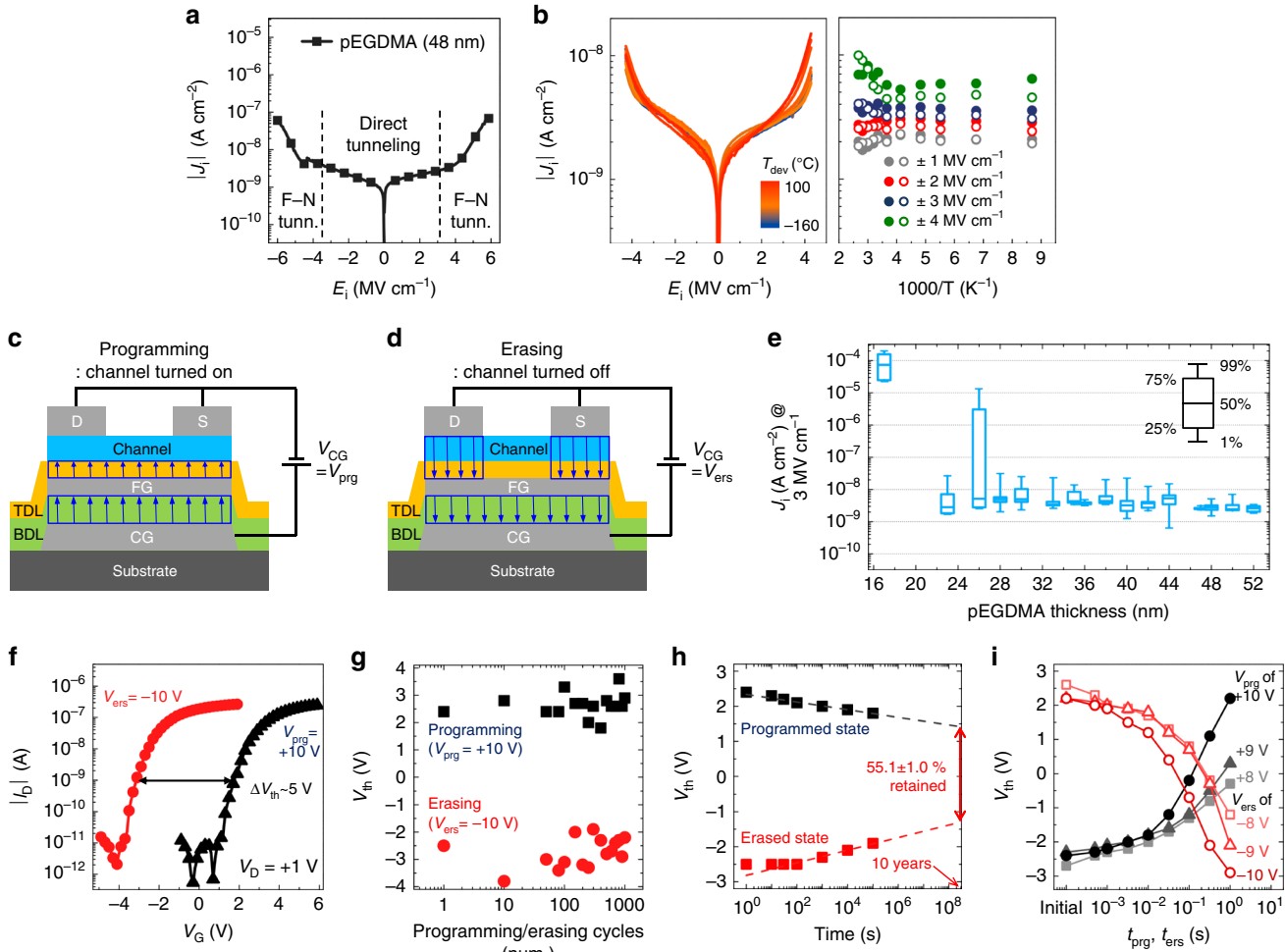

**Fig. 2** Properties of iCVD-processed polymer dielectric layers and an organic flash memory made thereof on plastic substrates. **a** $|J_i|$ vs. $E_i$ characteristics of a Al/pEGDMA (48 nm)/Al device. The *vertical, dashed lines* are a guide to eyes that divides low-field, direct-like tunneling regime and high-field, Fowler–Nordheim (F–N)-like tunneling regime. The latter can be easily distinguished from the low-field regime due to the clear field-enhanced $J_i$ over $E_i$. In both regimes, it is likely that each tunneling mechanism is coupled with trap-assisted tunneling (TAT)[57, 58]. **b** $|J_i|$–$E_i$ characteristics of Al/pEGDMA (36 nm)/Al devices measured at various device temperatures ($T_{dev}$) (*left*) and Arrhenius plots (*right*) transformed from them. **c**, **d** Sketches of electric field (shown as *arrows*) distribution inside flash memory under programming (**c**) and erasing (**d**) operations. **e** Statistical distribution of $J_i$ at $E_i$ of 3 MV cm$^{-1}$ obtained for the Al/pEGDMA/Al devices with various pEGDMA thicknesses. **f–i** Device characteristics of the proposed flash memories fabricated on a PET substrate: transfer memory characteristics (**f**), endurance test result over repeated programming/erasing operations (**g**), data retention characteristics (**h**), and memory speed characteristics for various programming/erasing voltages (**i**). All the transistor and memory characteristics in Fig. 2 and those appearing later have been measured under controlled atmosphere in an N$_2$-filled glove box, unless specified otherwise

of the bilayer dielectric structure, which is the key to flash memory operations. All of these considerations have made it highly challenging to realize true flash memories in flexible form factors.

Herein, we try to overcome these hurdles and realize highly flexible flash memories by employing thin polymeric insulators grown with initiated chemical vapor deposition (iCVD), a vapor-phase growth technique for polymers pioneered by Gleason and her coworkers[51]. With tunneling-limited characteristics, these dielectrics have recently been demonstrated as flexible gate insulators that are important for low-voltage flexible TFTs[52]. In this work, it is further shown that the iCVD-based polymeric insulators can make a significant contribution to flash memories as well. With a rational design and material choice that carefully consider both applied and built-in electric fields across the dielectric layers for each operating condition, the proposed approach simultaneously enables long retention (projected to be ~>10 years) and low-programming/erasing voltages comparable to the present

industrial standards (~±10 V). We then take full advantage of the mild, solvent-free, and low-temperature growth of iCVD to demonstrate flexible flash memories on various non-rigid substrates including ultrathin plastics and paper.

## Results

**Design and characterization of organic flash memories.** The proposed non-volatile memory devices are based on C$_{60}$-based organic TFTs, wherein the gate insulator is replaced with a TDL/FG/BDL, as shown in the schematic device structure presented in Fig. 1a. C$_{60}$ was chosen mainly for its consistency that led to a mobility larger than 1 cm$^2$ V$^{-1}$ s$^{-1}$ with a good run-to-run reproducibility[53–56]. Two iCVD processed polymer films of poly(1,3,5-trimethyl-1,3,5-trivinyl cyclotrisiloxane) (pV3D3) and poly(ethylene glycol dimethacrylate) (pEGDMA)[52], whose chemical structures are shown in Fig. 1c, are employed for TDL and BDL, respectively. While pV3D3 was chosen for its excellent insulating properties and down scalability that had

previously been reported by Moon et al.[52], pEGDMA was selected not only for its consistent insulating performance comparable to that of pV3D3, but also for its dielectric constant larger than that of pV3D3, which will be discussed later in details. As can be seen in the false-color high-resolution transmission electron microscopy (HRTEM) image in Fig. 1a, these polymers make a conformal coating on the bumpy Al surfaces used as a CG and a FG, respectively, thanks to the characteristics inherent to the vapor-based iCVD method. This conformal growth is highly beneficial in the present case because it can minimize a chance of forming accidental electrical short paths. Like pV3D3[52], pEGDMA polymer layers also have superb insulating properties even on flexible substrates, with low leakage current density ($J_i$) and a relatively high breakdown field of over 6 MV cm$^{-1}$ (Fig. 2a). This is due not only to the conformal coverage but also to the tunneling-limited insulating properties of iCVD-based polymers, which are characteristic to near-ideal dielectric layers with low trap densities[52]. The tunneling-limited behavior can be verified from their $J_i$ vs. applied electric field ($E_i$) characteristics, which are almost independent of the device temperature over a wide range, as shown in ref. [51] for pV3D3 and in Fig. 2b for pEGDMA.

From the viewpoint of memory operation, the $J_i$–$E_i$ characteristics of a metal-insulator-metal (MIM) device with near-ideal dielectric layers may be divided into two distinctive regions as shown for pEGDMA in Fig. 2a: (i) a low-$E_i$ regime where $J_i$ is significantly low due to direct-like tunneling where tunneling should be made through a full barrier width; and (ii) a high-$E_i$ regime where $J_i$ exhibits field-enhanced behavior, due to Fowler–Nordheim (F–N)-like tunneling, which involves tunneling through field-induced triangular barriers. It should be noted that, in both regimes, each tunneling mechanism could be further assisted with trap-assisted tunneling (TAT) found in dielectrics like high-quality Al$_2$O$_3$[57, 58], where a localized trap can help the respective tunneling processes by playing a role like that of a "stepping stone." Staying in the low-$E_i$ region is important in terms of memory retention, while utilizing field-enhanced, F–N-like tunneling can become critical for programming and erasing. Hence, it is crucial to ensure the field across TDL (=$E_{TDL}$) is high enough during programming/erasing so that $J_i$ is dominated by F–N-like tunneling. On the other hand, the field across BDL (=$E_{BDL}$) should always remain low to prevent charges from flowing through BDL in all the cases and to let charges flow into or out from FG only through TDL, if necessary for programming or erasing. To fulfill these seemingly conflicting conditions simultaneously, it is necessary to first look at the voltages across BDL (=$V_{BDL}$) and TDL (=$V_{TDL}$) and the associated $E_{BDL}$ and $E_{TDL}$ when a bias (=$V_{CG}$) is applied to a CG with respect to a source (S) and drain (D) electrodes. $E_{BDL}$ and $E_{TDL}$ are linked to each other via the so-called coupling ratio ($\alpha_{CR} = V_{FG}/V_{CG}$) and thus depend on the properties of both BDL and TDL, rather than solely on the properties of their own dielectric layers. Consider, for simplicity's sake, an S/TDL/FG/BDL/CG structure, which is equivalent to the memory structure underneath S electrode except for the absence of a semiconductor layer. Under capacitive approximation and in the absence of a built-in field, it can be shown that $E_{TDL}/E_{BDL}$ scales with $k_{BDL}A_{BDL}/(k_{TDL}A_{TDL})$, where $k_i$ and $A_i$ ($i$ = BDL or TDL) are the relative dielectric constant and area of the insulating layers under study (see Supplementary Fig. 1 and Supplementary Note 1 for further details). pEGDMA used as a BDL in this work was found to have a $k$ value of 3.0[59], which is significantly higher than that of pV3D3 used as a TDL (=2.2). This tends to make it easy for the condition of $E_{TDL} > E_{BDL}$ to be met at a given $V_{CG}$ for programming/erasing, and thus provides a foundation to realize the required asymmetry between carrier flows though BDL and

TDL. It is noteworthy that the dielectric constant of parylene, a polymer produced from vapor phase and commonly used in many areas, is on par with that of pEGDMA. Hence parylene could also be used for a BDL in this scheme. One concern for parylene is, however, its relatively low break-down field[60]. Nevertheless, the report by Song et al. demonstrates that the dielectric properties of polymers could be significantly improved upon use of soft contacts such as transferred graphene[61]. If the break-down field of parylene can further be increased in this way, parylene may also be used as one of the gate dielectric layers for memory devices.

Once $k_{TDL}$ and $k_{BDL}$ are chosen, one needs to determine the thicknesses of BDL (=$d_{BDL}$) and TDL (=$d_{TDL}$) that can fulfill the field requirements for all the memory operation conditions. In the real memory structure (instead of the aforementioned simplified case), the detailed energy band alignment should be carefully taken into account, because it can lead to built-in fields across BDL and TDL and thus exert an influence on the net field across those layers. It should further be noted that the built-in fields vary depending on the memory state because they are influenced by the differential increase or decrease in the amount of charge confined within FG (see Supplementary Table 1 and description therein for further details). Another complication, which results from the inherent properties of a typical organic TFT, is the change in $\alpha_{CR}$ associated with the dependence of the capacitance of the FG/TDL/channel/S(D) structure (=$C_{TDL}$) on the operation modes. Due to the gate-modulated conductivity of the channel, the effective thickness and area determining $C_{TDL}$ during programming (i.e., channel is ON) are $d_{TDL}$ and the area underneath the channel that overlaps with FG, respectively; in contrast, those relevant to the erasing process (i.e., channel is OFF) are $d_{TDL}$ plus the channel thickness ($d_{channel}$) and the area underneath D and S electrodes overlapping with FG, respectively (Fig. 2c, d). The energy diagrams in Fig. 1b were constructed by considering the location of Fermi level for C$_{60}$ and its interfacial alignment with Ca[62], the spatial distribution of the vacuum level shift associated with the confined charges, the applied $V_{CG}$ (if any), and operation-mode-dependent $\alpha_{CR}$, etc. With a BDL thicker than a TDL (in this work, $d_{BDL}/d_{TDL}$ = 1.8–5), energy diagrams are configured in such a way that F–N-like tunneling can occur efficiently across TDL, while it is much less effective for BDL due either to the low net electric field (during the programming operation; Fig. 1b "programming") or to the energy barrier configuration, which is relevant to direct tunneling instead of F–N tunneling (during the erasing operation; Fig. 1b "erasing").

When choosing $d_{TDL}$ and $d_{BDL}$, it should also be kept in mind that these values should be larger than the critical thickness (=$d_{crit}$) of the corresponding dielectric layer, below which the layer's insulating properties will quickly be degraded[47, 63]. A statistical study done for pEGDMA-based MIM devices fabricated in several batches indicates that $d_{crit}$ of pEGDMA is ca. 30 nm (Fig. 2e). We have thus adopted the pEGDMA layer that is at least thicker than ca. 30-40 nm. With $d_{crit}$ of pV3D3 being ca. 10 nm[52], the total dielectric thickness can be maintained within 50–68 nm. With this total thickness, we were able to keep the programming and erasing voltages ($V_{prg}$ and $V_{ers}$) relatively low while still fulfilling all the requirements mentioned above. For example, the proposed flexible organic flash memory with 40 nm-thick pEGDMA and 16 nm-thick pV3D3 on PET substrates exhibited transistor characteristics that can be switched on at less than 3 V; the device also showed a sufficient voltage shift ($\Delta V_{th}$) of ~5 V with $V_{prg}$ and $V_{ers}$ as low as $\pm 10$ V (Fig. 2f). Note that these levels of $\Delta V_{th}$ and programming/erasing voltages are comparable to those of conventional Si-based flash memory devices[2, 64]. With $|\Delta n_{fg}| = C_{BDL} \times |\Delta V_{th}|/e$, wherein $\Delta n_{fg}$ and $e$

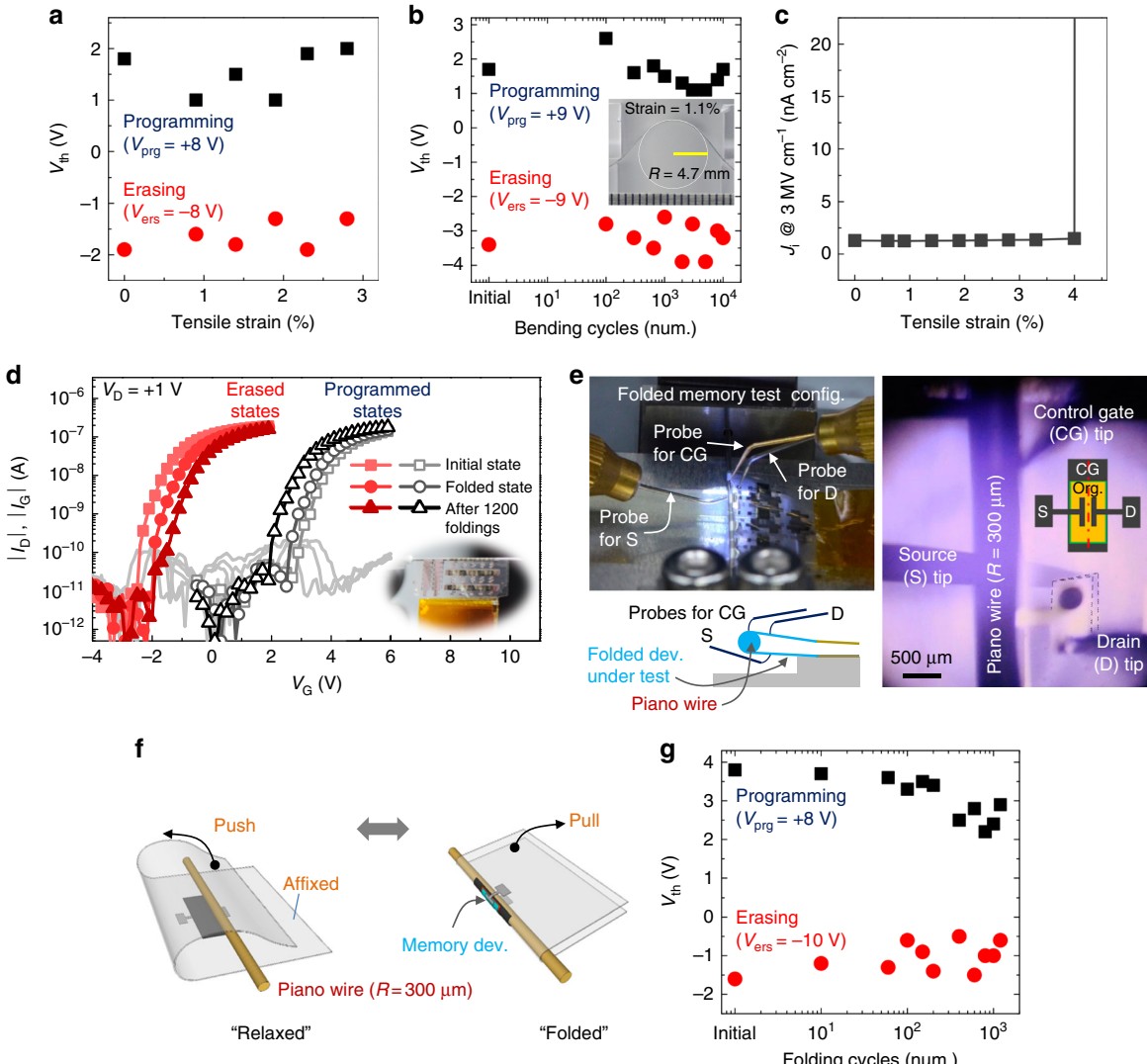

**Fig. 3** Characteristics of the proposed flexible flash memory under or after mechanical deformation. **a**, **b** $V_{th}$ measured upon programming/erasing operations for devices under various bending induced tensile strains (**a**), and $V_{th}$ measured after repetitive bending cycles with tensile strain of 1.1% (**b**), (*inset*: a photograph of the flexible organic flash memory fabricated on 100 μm-thick PET films). **c** $J_i$ values of Al/pEGDMA (40 nm)/Al devices at an $E_i$ of 3 MV cm$^{-1}$ measured as a function of various tensile strains. **d** Transfer memory characteristics of the proposed memory devices fabricated on 6 μm-thick Mylar$^{TM}$ substrates ("ultrathin Mylar" hereafter) in their flat and folded states, and after 1200-time folding cycles. Folding radius is 300 μm and defined by the radius of a piano wire. *Inset* shows a photograph of the folded flash memory fabricated on the Mylar$^{TM}$ film. **e** Photograph of a memory device on the ultrathin Mylar being characterized under the folded state. A top-view microscope image and a side-view probing diagram are also presented to show that folding-induced strain is applied exactly along the channel parallel to the long edges of S and D electrodes as shown in the schematic diagram presented in the *inset*. Such alignment can be witnessed by the contact pads (shown as areas surrounded by the *dashed lines*) of S and D that are overlapped each other. Note that the probe for S makes a contact from underneath the folded device. **f** A schematic diagram showing the method for alternating between "relaxed" and "folded" states for measurement of memory characteristics after repeated folding cycles. **g** $V_{th}$ of the memory on ultrathin Mylar substrates measued upon programming/erasing operations after repeated folding cycles at 300 μm bending radius vs. the number of folding cycles

refer to the change in carrier density stored in FG and charge of an electron, respectively, the observed $\Delta V_{th}$ corresponds to $\Delta n_{fg}$ of $2.1 \times 10^{12}$ cm$^{-2}$, which is on par with those of most flash-like memories found in the literature (see Supplementary Table 2). The reason why $\Delta n_{fg}$ exhibits little variation among different devices, albeit their widespread $\Delta V_{th}$, is that a flash memory with lower $C_{BDL}$ generally requires higher operating voltages and thus demands higher $\Delta V_{th}$ for proper memory operation, making the $C_{BDL} \Delta V_{th}$ product (and thus $\Delta n_{fg}$) remain in the similar range among different systems. In addition, the flexible memory devices in this study showed reliable programming and erasing behaviors for repetitive operations of 1000 times (Fig. 2g and Supplementary Fig. 2).

Moreover, the proposed memory devices exhibited significantly long retention time ($t_{ret}$), estimated to be $3.2 \times 10^8$ s ($\approx$10 years), during which over 50% of the initial $\Delta V_{th}$ is projected to be preserved (Fig. 2h). As can be seen in the energy diagram given in Fig. 1b) for "Programmed state", presence of the stored charges in FG tends to yield band bending in energy bands for both TDL and BDL. Because their slopes are still relatively small, however, the field across the insulators remain within the low-field regime. In this case, tunneling probability can be maintained low enough to enable long retention, provided that there are no other leakage paths such as electrical shorts. The observed level of large $t_{ret}$ is thus consistent with tunneling-limited insulating properties of pV3D3 and pEGDMA used in this work. Nonetheless, one should

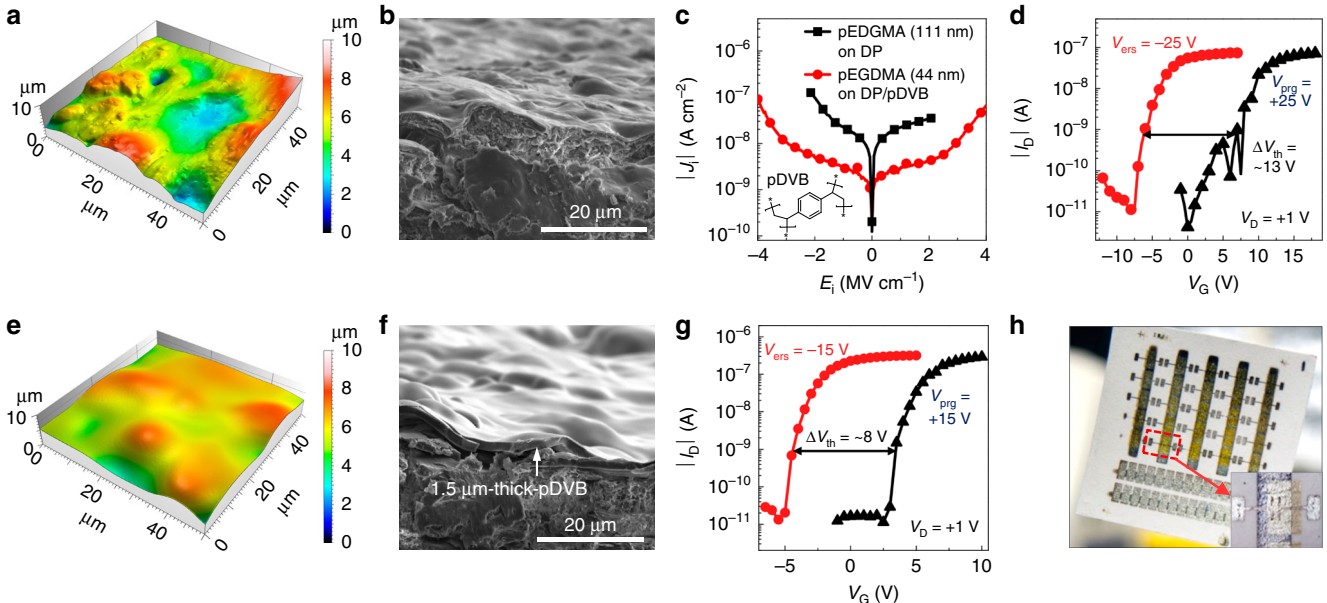

**Fig. 4** Organic flash memories on papers for disposable electronics. **a, b** Surface and cross-sectional morphologies of a pristine dye-sublimation paper (DP) characterized with atomic force microscopy (AFM) and scanning electron microscopy (SEM). **c** $J_i$–$E_i$ characteristics of pEGDMA layers measured from Al/pEGDMA (111 nm)/Al devices fabricated on a pristine DP (*black line* with *squares*) and that of Al/pEGDMA (44 nm)/Al on a pDVB-coated DP. The molecular structure of pDVB is shown in the *inset*. **d** Transfer memory characteristics of a flash memory fabricated on a pristine DP ($d_{BDL} = 111$ nm). **e, f** Surface and cross-sectional morphologies of the pDVB-coated DP. **g** Transfer memory characteristics of a flash memory fabricated on a pDVB-covered DP ($d_{BDL} = 44$ nm). **h** A photograph of the proposed paper-based organic flash memories

keep in mind that the quoted retention time is the extrapolated value that considers only the electrical aspects and that the actual memory retention is subject to degradation depending on ambient conditions. The long-term test with proper encapsulation should be made to fully assess their actual retention time in practice. As for programming/erasing speed, the device showed a $\Delta V_{th}$ of ca. 2 V at $t_{prg}$ and $t_{ers}$ as low as 10 ms (Fig. 2i), simultaneously fulfilling all the requirements for key performance parameters—reasonably low voltage and short time span for programming and erasing; and sufficiently long retention.

**Mechanical flexibility of the proposed organic flash memories**. To estimate the mechanical durability of the proposed organic flash memory, a device fabricated on a 250 µm-thick PET substrate was tested as a function of flexural tensile strain by adjusting the bending radius of curvature. The device showed consistent programming/erasing behaviors after flexural deformation corresponding to tensile strain of up to 2.8% (Fig. 3a and Supplementary Fig. 3), and it still exhibited memory characteristics even after the 10,000 bending cycles at a flexural strain of 1.1% (Fig. 3b and Supplementary Fig. 4). This excellent mechanical durability against flexural strain can be attributed to the highly flexible nature of the iCVD-based polymeric dielectric layers. Both the pEGDMA and pV3D3 dielectric layers maintained their insulating property at tensile strain of up to 4% in a bending test using MIM devices (Fig. 3c, Supplementary Fig. 3, and refs. [51, 52]. these results are in sharp contrast to the cases of inorganic dielectric layers that typically lose their insulating property with strain of ~1% in a similar test[37].

It is noteworthy that the onset strain for failure (=$\varepsilon_{onset}$) is lower for flexible memory devices than for the corresponding flexible MIM devices. This is attributed to the relatively complex device structure of memory devices, which involves a larger number of heterogeneous interfaces than is present in MIM devices. This is consistent with the $\varepsilon_{onset}$ value of 3.3% obtained for a $C_{60}$-based TFT transistor having the same configuration as

the proposed flash memory except that Al FG is not included (see Supplementary Fig. 5).

Even with a value of $\varepsilon_{onset}$ smaller than those of the corresponding TFTs or MIM devices, the observed value is still unprecedented for flexible flash memories[13–18] and is large enough to realize ultra-flexible form factors with a thin plastic substrate[10, 37, 39]. This will allow for a greater degree of freedom in designing flexible electronic systems, e.g., for wearable or skin-attachable electronic products. To verify this, the memory devices under study were fabricated on 6 µm-thick Mylar^TM films using 33 nm-thick pEGDMA as a BDL and 17 nm-thick pV3D3 as a TDL. As can be seen in Fig. 3d, the devices were easily bendable over a needle with the radius of merely 300 µm. The resultant devices show non-volatile memory operation comparable to that of the devices prepared on thick PET substrates. As the 6 µm-thick plastic substrate significantly reduces the bending-induced strain at its surface, the achievable bending radius can become so small that devices on top of it can almost be folded. When tested with a custom-designed bending machine that can define a sub-mm bending radius with the assistance of a rigid piano wire (Fig. 3e, f, Supplementary Fig. 6, and Supplementary Movies 1 and 2), these devices were shown to maintain their programming and erasing capabilities in the folded state as well as after 1200-time folding cycles (Fig. 3d, g) for a bending radius of 300 µm. This demonstrates their potential as nearly foldable and light-weight flash memory devices, which will be pivotal for epidermal or imperceptible electronics[10, 37, 39]. The possibility of fabricating these devices on Mylar^TM or PET substrates is a direct benefit of the low-temperature processability of the iCVD process, as these polyester films have relatively low-service temperature and glass-rubber transition temperature. Upon comparison with the previous works, the result shown here can be regarded as a significant breakthrough toward realization of highly flexible flash memory that satisfies both low $V_{prg}$, $V_{ers}$, and long $t_{ret}$ (Supplementary Table 2 and Supplementary Fig. 7).

**Realization of flash memories on papers for disposable electronics**. Another advantage of the iCVD process that can be useful for flexible memory devices is its solvent-free nature[52, 65, 66]. Together with the conformal coating capability and low-temperature processability, this benefit provides the potential to build memory devices on a wide range of unconventional substrates, including those with a low damage threshold to liquid as well as those with relatively large roughness[52, 65, 66]. In this respect, one attractive yet highly challenging substrate to build memory devices on is paper, which has all the unfavorable conditions mentioned above[67–79]. To determine whether substrates for the proposed memory can be extended even to paper, we built our devices on so-called dye-sublimation papers (DP), which has a significantly rough surface with bumps in both micro- and nano-scale (Fig. 4a, b and Supplementary Fig. 8). Note that DP does not have a plastic coating unlike the thick specialty photo paper used for inkjet printing of photographs. It was found that, thanks to the conformal coating capability of iCVD, a 111 nm-thick pEGDMA layer exhibited a considerable insulating property even on such a rough DP surface, as shown as a *black line* with *squares* in Fig. 4c. An organic flash memory fabricated on the DP surface with the 111 nm-thick pEGDMA-based BDL and a 44 nm-thick pV3D3-based TDL was shown to be successfully operable with $\Delta V_{th}$ of ~13 V for $V_{prg}$ or $V_{ers}$ of ±25 V (Fig. 4d).

To further drive down the operation voltage of memory devices on paper, we used the iCVD process to apply a 1.5 µm-thick poly (divinylbenzene) (pDVB) layer on the pristine DP surface so that nano-scale bumps were smoothened and only the relatively large micron-scale profiles remained. pDVB was chosen mainly for its relatively high deposition rate (~µm h$^{-1}$) and, moreover, for its planarizing capability. (Fig. 4e, f and Supplementary Fig. 8) In this case, even a 44 nm-thick pEGDMA layer showed a sufficiently good insulating property that was on par with the property observed in devices on glass or plastic substrates, as can be seen as a *red line* with *circles* in Fig. 4c. With this modification, the organic flash memory with the 44 nm-thick pEGDMA BDL and a 24 nm-thick pV3D3 TDL showed non-volatile memory operation with $\Delta V_{th}$ of ca. 8 V at $V_{prg}$ or $V_{ers}$ of ±15 V (Fig. 4g, h). To the best of our knowledge, this is the first demonstration of the flash memories on paper with practically meaningful operation voltage and a sufficient memory window. The results clearly demonstrate the immense potential of the iCVD process for flash memories on almost any kinds of flexible substrates.

## Discussion

In summary, flexible organic flash memory devices realized in this work on various substrates exhibit both industry-compatible operating voltage (~10 V) and relatively long-projected retention time. Thanks to the virtue of iCVD-based polymeric insulating layers with a near-ideal insulating property limited by carrier tunneling, an organic flash memory can be designed, in a systematic fashion, based on a TDL/FG/BDL structure. The versatile synthetic capability of the iCVD process makes it possible to choose, from the pool of various iCVD-based polymers, a proper insulator that can meet the conditions that are key to the systematic design (e.g., a TDL with low $d_{crit}$; a BDL with a higher dielectric constant than a TDL). Furthermore, the advantages of the iCVD process, such as low process temperature, solution-free nature, and highly conformal deposition, help one to realize flash memories on various flexible substrates such as PET, ultrathin Mylar$^{TM}$, and paper, which would otherwise be highly challenging to build memory devices on. The proposed devices on thick PET substrates are found to be able to withstand flexural strain of up to 2.8%, which is unprecedented

for non-volatile memory devices. It is noteworthy that the proposed approach is not specific to $C_{60}$, which is known to be air-sensitive and thus subject to degradation without encapsulation; some of the air-stable compounds[80, 81] can also be used provided that their preparation is compatible with flexible substrates of interest. All these advantages are expected to allow non-volatile memories to be fabricated on virtually any flexible substrates while maintaining a sufficient level of performance as well as a high degree of mechanical flexibility. We believe the proposed approach will open a path for forthcoming electronics that are expected to be ubiquitous in daily life by providing an effective means for wearable, foldable, or disposable electronics.

## Methods

**Initiated chemical vapor deposition process**. Each of the monomers called ethylene glycol dimethacrylate (EGDMA), 1,3,5-trimethyl-1,3,5-trivinyl cyclotrisilo-xane (V3D3), and divinylbenzene (DVB) was vaporized and delivered to a custom-built iCVD reactor with the initiator, *tert*-butyl peroxide (TBPO) for the formation of pEGDMA, pV3D3, and pDVB, respectively, via free-radical polymerization reaction. The growth rates of pV3D3, pEGDMA, and pDVB were 1.5, 5, and 15 nm min$^{-1}$, respectively. The process pressure was maintained at 70–300 mTorr, and the filament was heated to 160–200 °C. The substrate temperature was maintained at 40 °C. The thickness of the deposited polymer films was controlled in situ by a He–Ne laser (JDS Uniphase) interferometer system.

**Surface and film characterization**. To observe the cross-sectional structure of the non-volatile organic flash memory devices, the devices were sliced using a dual-beam focused ion beam (FIB, Helios Nanolab 450). HRTEM (Titan cubed G2 60-300) images were then obtained. The surface morphology of the bare dye-sublimation paper (DP) and the DP coated with pDVB was characterized using AFM (Nanosurf Nanite AFM & C3000 controller).

**Device fabrication**. The glass substrates were cleaned with detergent dissolved in deionized (DI) water, DI water, acetone, and isopropyl alcohol (IPA) successively in an ultrasonic bath for 20 min each, and dried at 100 °C for 12 h in a vacuum oven. PET substrates were cleaned with IPA. Ultra-thin Mylar and DP substrates were not cleaned with solvent; instead, a stream of compressed N$_2$ gas was just blown onto the substrates before fabrication.

For the planarization, PET and Mylar substrates were treated with air plasma (CUTE-1MP, Femto Science Inc.) for 1 min at 70 W. Directly after the plasma treatment, PET and Mylar films were spin-coated with a transparent photoresist (SU-8, Kayaku Microchem) at 2000 rpm (30 s).

For the fabrication of the bottom-gated $C_{60}$-based non-volatile flash memory, 100 nm-thick Al as CG was deposited by thermal evaporation with a base pressure lower than 10$^{-6}$ Torr using shadow masks with a deposition rate of ~0.1 nm s$^{-1}$, and pEGDMA polymer insulators as the BDL were deposited by the iCVD process on the top of CG electrodes with the desired thicknesses. The 15 or 50 nm-thick Al as the FG was then thermally evaporated in the same manner as Al prepared for the CG. pV3D3 polymer insulators as TDL were then deposited by the iCVD process on top of the FG electrodes. The memory devices were then completed by sequential thermal evaporation of 50 nm-thick $C_{60}$ channel layers (~0.07 nm s$^{-1}$) and Ca (20 nm)/Al(100 nm) layers as source (S) and drain (D) electrodes. Patterns for electrodes were defined with shadow masks.

**Electrical characterization of devices**. All current density vs. voltage characteristics ($J$–$V$) were analyzed using a semiconductor parameter analyzer (HP4155A or HP4156C, Agilent). Electrical characteristics of any devices with $C_{60}$ were measured under controlled atmosphere in an N$_2$-filled glove box unless noted otherwise. Capacitance vs. voltage characteristics ($C$–$V$) of MIM devices were measured using a precision LCR meter (HP4284A, Agilent). Dielectric constants ($k$) were estimated by using the relationship among $C$, $k$, and thickness of an insulator, following the procedure described in "Methods" section in the previous work by Moon et al.[52]. To measure the $J$–$V$ characteristics of the MIM devices with various values of device temperature ($T_{dev}$), the samples were placed on a hot chuck in a customized vacuum chamber with probe tips, and $T_{dev}$ was systematically controlled by a combination of liquid N$_2$ gas flow and the chuck heating. To evaluate the electrical properties of the flexible (or foldable) MIM and non-volatile memory devices as a function of the bending radius and the number of bending cycles, the devices were tested with a custom-built bending tester. In case of a sub-mm bending radius, a piano-wire was employed and held tightly to maintain a proper tension for straightness. For nearly foldable devices on Mylar$^{TM}$ substrates, the center of the bending was carefully aligned to the center of individual MIMs and memory devices by using a microscope.

**Data availability**. All relevant data supporting the findings of this study are included within the paper and its Supplementary Information File, or available from the corresponding authors on reasonable request.

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

## Acknowledgements

This work was supported by Samsung Research Funding Center of Samsung Electronics under Project Number SRFC-MA1402-04. H.M. is grateful to Brain Korea 21 Plus Program for post-doctoral fellowship.

## Author contributions

S.L., H.S., S.G.I., H.M., and S.Y. designed the experiments on the use of iCVD-based polymers as dielectrics for non-volatile organic flash memory devices. S.L. carried out fabrication and characterization of organic flash memory devices on plastic substrates as well as on paper. H.S and S.G.I. carried out process design, fabrication, and characterization of iCVD-processed polymer films. S.L., H.M., and S.Y. worked on the design and electrical analysis of a device structure and electrical analysis. S.L., H.M., and S.Y. wrote the manuscript. All authors read and commented on the manuscript. H.M. and S.Y. contributed equally as corresponding authors.

## Additional information

**Competing interests:** The authors declare no competing financial interests.

