## [Peer Review File · Nature Communications]

Reviewers' comments:

Reviewer #2 (Remarks to the Author):

The authors report flexible organic flash memory fabricated on various soft substrates including plastic and paper. Upon optimizing the deposition process and the film thickness of two polymer dielectrics, the resultant devices showed good memory characteristics at low operation voltage of 10 V. Particularly, the memory devices exhibited long projected retention time up to 10 years and at the same time maintained excellent robustness in bending test at a bending radius down to 300 micrometers. The all demonstrated the great potential of these flexible flash memory devices for applications to foldable and disposable electronics. Despite that, however, as the key enablers of these memory devices, the two polymer dielectric layers (TDL and BDL) fabricated by iCVD technique have been published before in Ref. 51. In addition, the device fabrication, characterizations, and analyses reported here are quite conventional with respect to the literature. Therefore, this paper is not original enough to guarantee publication in Nature Communications.

Due to the reason, I believe the paper should be submitted to elsewhere.

Reviewer #3 (Remarks to the Author):

The manuscript by Seunghyup Yoo, Hanul Moon and their co-workers report several advances made in a large programme of work to lay down foundations for ultraflexible organic flash memory devices. The key advance here is the exploitation of the initiated CVD-deposited polymers, as exemplified by a vinyl-substituted cyclotrisiloxane and a dimethacrylate, to provide conformal pinhole-free dielectric layers at the 30-50 nm and 15-20 nm thickness range, respectively, needed to construct floating-gate organic flash memory devices by sequential evaporation. They achieved interesting performance levels (projected retention time, 1 y; programming voltage range, within +/- 10 V; max strain, 2.8%; etc). The authors have demonstrated similar levels also on paper. This shows the intrinsic resilience of organic polymers to bending stresses.

Although the achievements here are largely engineering, leveraging on dielectric materials identification reported by the same authors in Nature Materials (2015), and do not address critical integration questions (e.g. it is not known whether the iCVD processed dielectric layers are compatible with lithography needed in device integration), I consider their currently reported state-of-the-art to be of sufficiently general interest to be published in Nature Communications.

Nevertheless, it would be very good if the authors could address the following minor issues:

(i) Would the authors please consider to make a comparison with the performance (dielectric and mechanical) of parylene, currently the most popular CVD-deposited dielectric material? Could the authors say something about the dielectric similarities and differences between EGDMA, V3D3, DVB and parylene. Furthermore, testing in the traditional evaporated MIM structures produces results that depend on the contacts and evaporated top metal, which may not reflect intrinsic properties of the dielectric (see e.g. Song et al, Nature Nanotechnol (2013) doi: 10.1038/NNANO.2013.63). The

significance of this is that if soft contacts (not evaporated metal contacts) are used, the dielectric thicknesses might be further down-scaled to produce even more better device performances!

(ii) Chemical structures in Figs 1c and d appear to be incomplete. The authors should consider to denote the polymerized vinyl groups by proper chemistry notation, i.e, -(xxx)-

(iii) Fig 2a: Is it really "direct tunneling" and Fowler Nordheim tunneling? Through a 20+ nm thick dielectric? Or is it transport through metal filamental (defect) structures?

(iv) Images in Fig 3 may need higher resolution and more annotation for the reader to understand what is being shown.

Reviewer #4 (Remarks to the Author):

The authors demonstrate the application of two ultrathin polymers that prepared by iCVD method as blocking and tunneling dielectric layers in organic memories and these memory devices can be constructed on various soft substrates. Thanks to the excellent insulating properties of these iCVD derived polymer dielectrics, the devices can be programmed/erased at a low voltage and present impressive memory window. In addition, these memories can be fold-able and they exhibit very good data retention capability. The manuscript is well organized and all the results are supported by proper discussion. The preparation of polymer is not new which has been reported by authors. However, they used those known polymers to prepare memory devices which is new. It would be better if the authors can make a revision as follows.

1. In Fig 2a, how did the authors determine the boundary of two regions of direct tunneling and F-N tunneling?
2. Several types of polymer dielectrics were reported in their previous paper (Nature Materials, 2015, 14, 628-635), why did the authors choose pEGDMA as blocking dielectric material in this work? The author could add some advantages of this polymer over others in Line 70, Page 1.
3. The charge storage density of Al floating gate used in this memory device should be calculated and a comparison between this planar Al and metal nanoparticles in other reports shall be provided?
4. In Fig 2g, although long data retention has been recorded, the mechanism of data loss should be explained.
5. If possible, it's better to give a cross-sectional image of the edge overlap of CG/BDL/FG/TDL/Channel/S(D) in the memory device as charge leakage may occur through these paths.
6. There is a typo in expression 2 in Fig S1, the VFG behind the third equal sign should be revised to VCG.
7. The 10 years retention time by extrapolation is not acceptable and no one knows what will happen for such a long time with varying temperature and humidity conditions. Therefore it is better to remove such arguments.

Reviewer #5 (Remarks to the Author):

This paper reports on organic flash memory devices based on organic thin-film transistors with a floating gate electrode sandwiched between two dielectric layers.

The properties of these two dielectric layers play an important role in the overall performance of the memory device both in terms of operating voltages and memory retention time.

The main novelty of this work is in the nature and fabrication of the two dielectric layers that are used in these memory devices, namely a tunneling dielectric layer and a charge-blocking layer. In this work the authors employ a deposition technique referred to as initiated chemical vapor deposition (iCVD) that is based on the co-deposition of monomers with an initiator to form conformal polymeric layers via free-radical polymerization reactions. This technique developed at MIT in the Gleason group is well-known and has been employed in various organic optoelectronic devices, including organic field-effect transistors (e.g. ref. 51), but appears to be used in the context of memory devices based on OTFTs with a floating gate electrode for the first time.

The amount of data is extensive and the work of excellent quality. The description of the experiments is detailed and the description of the operational principles of the device didactic.

The paper could be further improved if the following comments are addressed:

1) I find the use of the term “platforms” in the title misleading as it is usually used in the context of an ensemble of technologies that are combined. Here, it looks like the authors meant to say that they demonstrated memory devices on various substrates.

2) Likewise, it is questionable if one should refer to PET and paper substrates as “soft”. Flexible or bendable might be a better term.

3) The authors should give credit to the seminal work on iCVD by proper referencing, as this technique appears to play an important role in the presented studies.

4) On line 70, page 1, the authors should replace “molecular structures” with “chemical structures.”

5) How was the dielectric constant of the TDL determined?

6) The atmospheric testing conditions should be specified. The electrical properties of C60 are known to degrade in air. Authors should comment on the environmental stability of their devices.

The strength of this paper is that the overall performance of these devices is impressive as operating voltages similar to those used in Si-based flash memory are demonstrated. The devices yield also long memory retention times. These devices are demonstrated on paper.

A major weakness of this paper is that it fails to provide a clear benchmark with the prior art in organic memory devices with similar geometry. There is substantial prior art as listed by the authors (refs. 13-36) but a quantitative comparison is lacking. The latter makes the assessment of the novelty and significance of this work difficult. I believe that such a quantitative comparison is needed to justify publication in a journal like Nature Communications.

REVIEWERS' COMMENTS:

Reviewer #3 (Remarks to the Author):

I've originally already recommended publication of the manuscript by Seunghyup Yoo and co-workers with minor revisions. The manuscript discloses new impressive benchmarks reached by rational materials selection and device design for floating-gate memory applications. This will likely stimulate new waves of research in different directions. In the revised manuscript now, the authors have further refined their manuscript and clarified several technical issues. I have no other queries, and can therefore recommend publication in its present form.

Reviewer #4 (Remarks to the Author):

The authors revised the manuscript as per reviewers' recommendation therefore I recommend for publication.

Reviewer #5 (Remarks to the Author):

In their revised version, the authors have thoroughly addressed all of my comments and recommendations. The quality of the paper has improved significantly. It meets now, in my opinion, the high standards for publication in Nature Communications.

Response to Reviewer's Comments and Summary of Changes

We would like to thank the referees for their thoughtful comments and careful review of our manuscript. The reviewers kindly had several comments and suggestions for improvements. Responses to each of the comments are summarized as follows:

Reviewers' comments:

Reviewer #3:

The manuscript by Seunghyup Yoo, Hanul Moon and their co-workers report several advances made in a large programme of work to lay down foundations for ultraflexible organic flash memory devices. The key advance here is the exploitation of the initiated CVD-deposited polymers, as exemplified by a vinyl-substituted cyclotrisiloxane and a dimethacrylate, to provide conformal pinhole-free dielectric layers at the 30-50 nm and 15-20 nm thickness range, respectively, needed to construct floating-gate organic flash memory devices by sequential evaporation. They achieved interesting performance levels (projected retention time, 1 y; programming voltage range, within +/- 10 V; max strain, 2.8%; etc). The authors have demonstrated similar levels also on paper. This shows the intrinsic resilience of organic polymers to bending stresses.

Although the achievements here are largely engineering, leveraging on dielectric materials identification reported by the same authors in Nature Materials (2015), and do not address critical integration questions (e.g. it is not known whether the iCVD processed dielectric layers are compatible with lithography needed in device integration), I consider their currently reported state-of-the-art to be of sufficiently general interest to be published in Nature Communications.

Nevertheless, it would be very good if the authors could address the following minor issues:

1. Would the authors please consider to make a comparison with the performance (dielectric and mechanical) of parylene, currently the most popular CVD-deposited dielectric material? Could the authors say something about the dielectric similarities and differences between EGDMA, V3D3, DVB and parylene. Furthermore, testing in the traditional evaporated MIM structures produces results that depend on the contacts and evaporated top metal, which may not reflect intrinsic properties of the dielectric (see e.g. Song et al, Nature Nanotechnol (2013) doi: 10.1038/NNANO.2013.63). The significance of this is that if soft contacts (not evaporated metal contacts) are used, the dielectric thicknesses might be further down-scaled to produce even more better device performances!

(Ans)

We would like to thank Reviewer#3 for many insightful comments, which we believe will be of great help to improve the quality of our manuscript. In Comment #1, Reviewer #3 is first asking authors to compare iCVD-based polymers with parylene, which is also a vapor-based polymer that is more common than iCVD-based polymers. Parylene is indeed a very useful insulating or passivation material widely used in biology and MEMS for its mild deposition condition, solvent-free nature, and/or conformal coverage. Being polymers, parylene is expected to be highly flexible as well. In such respects, parylene and iCVD-based polymers share many common benefits. Recently, several groups have worked on using parylene as solvent-free polymeric gate insulating layers. Such efforts have recently led to low-voltage, flexible organic TFTs based on 18-nm-thick parylene layers. This is a very encouraging news in that a similar device architecture and design criteria may be utilized with parylene, which is widely available. Furthermore, its relative dielectric constant is ca 3.0, which is comparable to that of pEGDMA. One concern for parylene is, however, its relatively low break-down field, which is reportedly to be around 2.5 MV/cm.

(Kondo, M. et al. *Applied Physics Express* **9**, 061602 (2016)) In contrast, both pEGDMA and pV3D3 can withstand 6 MV/cm. Having the large break-down field is very important because flash-like memory devices require tunneling-based programming and erasing, which can involve much higher electric field across dielectric layers than in the case of simple TFTs. Nevertheless, the report by Song et al. demonstrates that intrinsic dielectric properties could be compromised by the nature of contacts formed between the evaporated metal and an insulator of interest and that the true dielectric properties may be realized with soft contacts such as transferred graphene. If the break-down field of parylene can further be increased in this way, parylene may also be used as one of the gate dielectric layers for the proposed memory devices. One should keep in mind that it is still important to use two different dielectric layers such that the dielectric constant of BDL may be larger than that of TDL. For example, parylene ($k = 3.0$) and pV3D3 ($k = 2.2$) may be considered as BDL and TDL, respectively, but not parylene and pEGDMA due to their similar dielectric constants.

pDVB was used to deposited 1.5 μm -thick planarization layers on a rough paper surface, because it was identified to exhibit relatively high deposition rate (ca. 1000 nm/hr) and, at the same time, show a planarizing capability. Table R1 shown below summarizes the properties of the iCVD-based polymers used in this work and parylene.

Table R1. Summary of polymer materials used in this work

Polymer	Main use	Dielectric const.	Down-scalability as GI	Break-down field
parylene	Passivation, gate insulator (GI)	3.0	ca. 18 nm	ca. 2.5 MV/cm
pV3D3	GI	2.2	ca. 10 nm	ca. 6.0 MV/cm
pEGDMA	GI	3.0	ca. 30 nm	ca. 6.0 MV/cm
pDVB	Planarization	-	-	-

(Revision made in response to Comment#1)

1) Page 3 Line 13 (Page and line numbers in the *revised* version, unless specified otherwise)

pEGDMA used as a BDL in this work was found to have a k value of 3.0,⁵² which is significantly higher than that of pV3D3 used as a TDL ($=2.2$). This tends to make it easy for the condition of $E_{TDL} > E_{BDL}$ to be met at a given V_{CG} for programming/erasing, and thus provides a foundation to realize the required asymmetry between carrier flows through BDL and TDL. It is noteworthy that the dielectric constant of parylene, a polymer produced from vapor phase and commonly used in many areas, is on par with that of pEGDMA. Hence parylene could also be used for a BDL in this scheme. One concern for parylene is, however, its relatively low break-down field.⁶⁰ Nevertheless, the report by Song et al. demonstrates that the dielectric properties of polymers could be significantly improved upon use of soft contacts such as transferred graphene.⁶¹ If the break-down field of parylene can further be increased in this way, parylene may also be used as one of the gate dielectric layers for memory devices.

2) Page 6 Line 46

To further drive down the operation voltage of memory devices on paper, we used the iCVD process to apply a 1.5 μm -thick poly(divinylbenzene) (pDVB) layer on the pristine DP surface so that nano-scale bumps were smoothed and only the relatively large micron-scale profiles remained. pDVB was chosen mainly for its relatively high deposition rate ($\sim \mu\text{m/hr}$) and, moreover, for its planarizing capability. (Fig. 4e,f, and Supplementary Fig. 8 & 9).

2. Chemical structures in Figs 1c and d appear to be incomplete. The authors should consider to denote the polymerized vinyl groups by proper chemistry notation, i.e. -(xxx)-

(Ans)

We thank Reviewer #3 for his or her attention to the details that we missed. We have added the full chemical names of pV3D3 and pEGDMA in the caption of Fig. 1 so that readers can learn them even without reading through the main manuscript.

Please note that iCVD processed polymers are not in linear shape but form a lump of monomers with omnidirectional bridges; hence, polymerized vinyl groups, denoted by “--]*” are placed at unfamiliar positions in chemical structures as given in Fig. R1 shown below.

Fig. R1. Chemical structure of pV3D3 and pEGDMA.

(Revision made in response to Comment#2)

1) Fig. 1 Caption

Figure 1 | Structure of thin-film transistor based organic flash memories and their operation mechanism. a, The schematic device structure ... as described in details in Supplementary Table 1. c, Chemical structures of the poly(1,3,5-trimethyl-1,3,5-trivinyl cyclotrisiloxane) (pV3D3) and poly(ethylene glycol dimethacrylate) (pEGDMA) polymers used as TDL and BDL, respectively.

*“pV3D3” and “pEGDMA” in Fig. 1c have also been updated in the format of “full names (abbreviation).”

3. Fig 2a: Is it really "direct tunneling" and Fowler Nordheim tunneling? Through a 20+ nm thick dielectric? Or is it transport through metal filamental (defect) structures?

(Ans)

We are grateful to Review#3 for his or her insightful comment that raises important issues regarding the mechanisms involved in conduction through insulating layers used in this work. We agree with Reviewer #3 that we should have paid more attention when using the term “direct” tunneling and “F-N” tunneling. In the original manuscript, we were trying to mean, by “direct” tunneling, tunneling of carriers with no or little help of electric field; by “F-N” tunneling, in contrast, we were trying to mean field-assisted tunneling of carriers. While it is indeed the case, the rigorous definition of direct and F-N tunneling involves idealized cases that assume well-defined energy band structures, defect-free interfaces, and absence of defects or traps.

We agree with Reviewer#3 that the current densities of MIM devices based on pV3D3 or p(EGDMA) is higher than those expected for pure direct or pure F-N tunneling in both low- and high-field regions. Among various conduction

mechanisms considered for insulating media, the one that is most likely for pV3D3 or p(EGDMA) is trap-assisted (direct or F-N) tunneling (TAT). As can be seen in Fig. R2 shown below, presence of traps can increase the net tunneling rate in both direct and F-N regimes by playing a role similar to that of a “stepping stone”. The trap-assisted tunneling (TAT) has been employed to describe the leakage characteristics of dielectrics such as Al_2O_3 and SiO_xN_y , which show the similar level of current densities with the iCVD-based dielectrics with comparable thickness. Note that TAT is still a tunneling-based conduction in its essence and therefore has little temperature dependence; this is consistent with J - V characteristics of both pV3D3 and p(EGDMA) measured over a wide range of device temperature (T_{dev}). The slight enhancement of J at high E with temperature for $T_{\text{dev}} > \text{ca. } 300 \text{ K}$ can be attributed presumably to the increased contribution from Poole-Frenkel (PF) conduction, which is helped by both thermal energy and high electric field.¹

Figure R2. Energy band diagrams for trap-assisted tunneling (TAT) between electrodes in (a) “direct” (or low-field) tunneling regime^{R1} and in (b) “F-N” (high-field or field-assisted) tunneling regime.^{R2}

Reviewer also asked whether the carrier transport in pV3D3 and p(EGDMA) occurs through formation of metal filament; we believe it is less likely to be a dominant mechanism because identical J - E curves consisting of low J (on the order of nA/cm^2) at low E and a relatively high J at high E fields are *repeatedly* measured within the field range used in this work. Filament formation in iCVD-based pV3D3 has been observed and utilized for resistive random access memory (ReRAM), but it occurred only when one of the electrode was Cu instead of Al,^{R3} indicating that metal filament formation process in pV3D3 occurs with a specific electrode like Cu, but not with Al.

[R1] M. Specht *et al.*, *Appl. Phys. Lett.* 84, 3076 (2014). [3] T. W. Hickmott, *J. Appl. Phys.* 97, 104505 (2005).

[R2] B. L. Yang *et al.*, *Microelectronics Reliability* 44, 709 (2004).

[R3] B. C. Jang *et al.*, *ACS Appl. Mater. Interfaces.* 8, 12951 (2016)

(Revision made in response to Comment#3)

1) Page 1, line 54

... With ~~ideal~~ tunneling-limited characteristics, these dielectrics have recently been demonstrated as soft gate insulators that are important for low voltage flexible TFTs. ...

2) Page 2, line 25

From the viewpoint of memory operation, the J - E characteristics of a metal-insulator-metal (MIM) device with near-ideal dielectric layers may be divided into two distinctive regions as shown for pEGDMA in Fig. 2a: (i) a low- E regime where J is significantly low due to direct-like tunneling where tunneling should be made through a full barrier width; and (ii) a high- E

regime where J_i exhibits field-enhanced behavior, due to Fowler-Nordheim(F-N)-like tunneling, which involves tunneling through field-induced triangular barriers. It should be noted that, in both regimes, each tunneling mechanism could be further assisted with trap-assisted tunneling (TAT) found in insulators like high-quality Al_2O_3 ,^{57,58} where a localized trap can help the respective tunneling processes by playing a role like that of a “stepping stone.” where J_i rapidly increases with E_i . Staying in the low- E_i region is important in terms of memory retention, while utilizing field-enhanced, F-N-like tunneling can become critical for low-voltage programming and erasing. Hence, it is crucial to ensure the field across TDL ($=E_{\text{TDL}}$) is high enough during programming/erasing so that J_i is dominated by F-N-like tunneling.

3) Figure 2 Caption

Figure 2 | Properties of iCVD processed ultrathin polymer dielectric layers and an organic flash memory made thereof on plastic substrates. a, $|J_i|$ versus E_i characteristics of a Al/pEGDMA(48nm)/Al device. The vertical, dashed lines are a guide to eyes that divides low-field, direct-like tunneling regime and high-field, Fowler-Nordheim (F-N)-like tunneling regime. The latter can be easily distinguished from the low-field regime due to the clear field-enhanced J_i over E_i . In both regimes, it is likely that each tunneling mechanism is coupled with trap-assisted tunneling (TAT).^{57,58} b, $|J_i|$ - E_i characteristics (left) of Al/pEGDMA(36nm)/Al devices measured at various device temperatures (T_{dev}) (left) ...

4) References added.

- 57 Specht, M., Stadel, M., Jakschik, S., Schroder, U. Transport mechanisms in atomic-layer-deposited Al_2O_3 dielectrics. *Applied Physics Letters* **84**, 3076 (2004).
- 58 Yang, B. L., Lai, P. T., Wong, H. Conduction mechanisms in MOS gate dielectric films. *Microelectronics Reliability* **44**, 709 (2004).

- Continued in the next pages. -

4. Images in Fig 3 may need higher resolution and more annotation for the reader to understand what is being shown.

(Ans)

The microscope image Fig. 3e appears blurred in some areas; we seek a generous understanding from Reviewer that this is difficult to avoid, even with a high-resolution version, because it involves 3D structures (e.g. probes) and folded transparent substrates. Instead, let us provide readers with (i) more annotations for the images in Fig. 3; and with (ii) the enlarged version of Fig.3e in the Supplementary Fig. 7, so that they can better understand what is being shown.

(Revision made in response to Comment#4)

1) Fig. 3 e-g and its caption (with the original Fig. 3g moved to Supplementary Info.):

Figure 3 | Characteristics of the proposed flexible flash memory under or after mechanical deformation. a,b, V_{th} measured ... **e,** Photograph of a memory device on the ultrathin Mylar being characterized under the folded state. A top-view microscope image and a side-view probing diagram are also presented to show that folding-induced strain is applied exactly along the channel parallel to the long edges of S and D electrodes as shown in the schematic diagram presented in the inset. Such alignment can be witnessed by the contact pads (shown as areas surrounded by the dashed lines) of S and D that are overlapped each other. Note that the probe for S makes a contact from underneath the folded device. **f,** A schematic diagram showing the method for alternating between “relaxed” and “folded” states for measurement of memory characteristics after repeated folding cycles. **g,** V_{th} of the memory on ultrathin Mylar substrates measured upon programming/erasing operations after repeated folding cycles shown in f at 300 μ m bending radius vs. the number of folding cycles. **g,** Comparison of the present work with the TFT-based non-volatile memory devices reported in the literature. The graph classifies the devices according to the kinds of BDL/TDL layers; inorganic or organic ones. Those with ‘e’ values correspond to the performance data of flexible memory devices. In those cases, ‘e’ refers to the maximum strain tested in the corresponding work.

2) Supplementary Fig. 7: The enlarged version of Fig. 3e has been added as Suppl. Fig. 7(e).

(Additional Comment) Although the achievements here are largely engineering, leveraging on dielectric materials identification reported by the same authors in Nature Materials (2015), and do not address critical integration questions (e.g. it is not known whether the iCVD processed dielectric layers are compatible with lithography needed in device integration), I consider their currently reported state-of-the-art to be of sufficiently general interest to be published in Nature Communications.

(Ans) Integration was not the focus in the present work, but we agree with Reviewer that compatibility with common lithographic techniques is of premium importance for iCVD-based polymers to be used in device integration beyond fabrication of stand-alone MIM devices or TFTs. One can be assured that iCVD-based polymers are readily compatible with lithographic techniques. Selective removal of iCVD-based polymers can be easily done by the following procedures: (i) coat a iCVD-based polymer layer with photoresist (PR); (ii) expose a part of the PR layer under UV via a photomask; (iii) develop so that a part of the iCVD layer that will be removed has no PR left; (iv) remove the non-covered iCVD layer by plasma; (v) strip the remaining PR layer. The iCVD-based polymers do not get dissolved by most solvents, making them easier to work with materials used in common photolithography. The example shown in Fig. R3 is a 10-by-10 transparent and flexible active-matrix OLED whose fabrication involved via formation and opening of an anode area based on photolithographic definition.

[Figure R3 was redacted here]

Reviewer #4:

The authors demonstrate the application of two ultrathin polymers that prepared by iCVD method as blocking and tunneling dielectric layers in organic memories and these memory devices can be constructed on various soft substrates. Thanks to the excellent insulating properties of these iCVD derived polymer dielectrics, the devices can be programmed/erased at a low voltage and present impressive memory window. In addition, these memories can be fold-able and they exhibit very good data retention capability. The manuscript is well organized and all the results are supported by proper discussion. The preparation of polymer is not new which has been reported by authors. However, they used those known polymers to prepare memory devices which is new. It would be better if the authors can make a revision as follows.

1. In Fig 2a, how did the authors determine the boundary of two regions of direct tunneling and F-N tunneling?

(Ans)

Direct tunneling and F-N tunneling differ in that the latter has a rather strong dependence on electric fields. In Fig. 2a, one can clearly identify a region ($E > \sim 3\text{MV/cm}$), wherein J - E graph shows a strong field dependence. Please note the vertical scale is given in a logarithmic scale.

(Revision made in response to Comment#1)

1) Figure 2 Caption

Figure 2 | Properties of iCVD processed ultrathin polymer dielectric layers and an organic flash memory made thereof on plastic substrates. a, $|J_i|$ versus E_i characteristics of a Al/pEGDMA(48nm)/Al device. The vertical, dashed lines are a guide to eyes that divides low-field, direct-like tunneling regime and high-field, Fowler-Nordheim (F-N)-like tunneling regime. The latter can be easily distinguished from the low-field regime due to the clear field-enhanced J_i over E_i . In both regimes, it is likely that each tunneling mechanism is coupled with trap-assisted tunneling (TAT).^{57,58} b, $|J_i|$ - E_i characteristics (left) of Al/pEGDMA(36nm)/Al devices measured at various device temperatures (T_{dev}) (left) ...

2. Several types of polymer dielectrics were reported in their previous paper (Nature Materials, 2015, 14, 628-635), why did the authors choose pEGDMA as blocking dielectric material in this work? The author could add some advantages of this polymer over others in Line 70, Page 1.

(Ans)

Although several iCVD-based polymers were shown in authors' previous work (Nature Materials, 2015, 14, 628-635), not all of them show the down-scalability comparable to that of pV3D3. That was mainly the reason to choose pV3D3 as a major dielectric of interest in that previous work. pEGDMA was the best dielectric material next to pV3D3 in terms of down-scalability. We looked for dielectric layers that can show a very low level of current flow at a thickness smaller than ca. 50 nm for a wide range of electric field. pEGDMA provided a very consistent insulating performance similar to that of pV3D3, as long as it is thicker than ca. 30 nm. (See Fig. S2 in Supporting Information) Most of all, it had a higher dielectric constant than pV3D3, which is important in ensuring the asymmetric dielectric properties between BDL and TDL. Its dielectric constant was the highest of all the polymers reported in our previous work. ($k_{\text{pEGDMA}}=3.0 > k_{\text{pPFDA}}=2.7 > k_{\text{pIBA}}=2.6 > k_{\text{pV3D3}}=2.2$)

In short, pEGDMA was chosen because it meets both of the requirements for (i) a relatively good down-scalability/consistent insulating performance and (ii) a dielectric constant that is higher than that of pV3D3.

(Revision made in response to Comment#2)

1) Page 1, Line 77: (Page and line numbers in the *revised* version, unless specified otherwise)

... Two iCVD processed polymer films of poly(1,3,5-trimethyl-1,3,5-trivinyl cyclotrisiloxane) (pV3D3) and poly(ethylene glycol dimethacrylate) (pEGDMA),⁵² whose ~~molecular~~ **chemical** structures are shown in Fig. 1c, are employed for TDL and BDL, respectively. **While pV3D3 was chosen for its excellent insulating properties and down-scalability that had previously been reported by Moon et al.,⁵¹ pEGDMA was selected not only for its consistent insulating performance comparable to that of pV3D3 but also for its dielectric constant larger than that of pV3D3, which will be discussed later in details.** As can be seen in the false-color high-resolution transmission electron microscopy (HRTEM) image in Fig. 1a, these polymers make a conformal coating on the bumpy Al surfaces used as a CG and a FG, respectively, thanks to the characteristics inherent to the vapor-based iCVD method. This conformal growth is highly beneficial in the present case because it can minimize a chance of forming accidental electrical short paths. Like pV3D3,⁵² pEGDMA polymer layers also have superb insulating properties even on flexible substrates, with low leakage current density (J_l) and a relatively high breakdown field of over 6 MV cm⁻¹ (Fig. 2a). ...

3. The charge storage density of Al floating gate used in this memory device should be calculated and a comparison between this planar Al and metal nanoparticles in other reports shall be provided.

(Ans)

The relation between the stored charge and carrier density in the floating gate (Q_{fg} and n_{fg} , respectively) and the amount of voltage shift (ΔV_{th}) in a flash memory is known to be:

$$|\Delta Q_{fg}| = e |\Delta n_{fg}| = |\Delta V_{th}| \times C_{BDL} \dots (1),$$

where C_{BDL} is the capacitance density of a CG/BDL/FG. With C_{BDL} of 66.3 nF·cm⁻², ΔV_{th} of about 5V obtained for the proposed organic flash memory in Fig. 2e corresponds to ΔQ_{fg} and Δn_{fg} of 3.4×10⁻⁷ C·cm⁻² and 2.1×10¹² cm⁻².

It is worthwhile to note that a larger or smaller Δn_{fg} may not necessarily be a criterion to assess the capability of a given flash-type nonvolatile memory. What counts more is how effectively a given memory device generates ΔV_{th} required for a sufficient contrast between '0' and '1' states. It should also be noted that the required ΔV_{th} can differ depending on the dielectric and charge storage structures of a given memory. Nevertheless, most of them, including those with metal nanoparticles replacing a floating gate, appear to have Δn_{fg} on the order of 10¹² cm⁻²; this is because a flash memory with lower C_{BDL} generally requires higher operating voltages and thus demands higher ΔV_{th} for proper memory operation. This opposing trends between C_{BDL} and ΔV_{th} tend to make the $C_{BDL}\Delta V_{th}$ product (and thus Δn_{fg}) remain in the similar range among different systems. Please see Supplementary Table 2 added in the revised Supplementary Information.

(Revision made in response to Comment#3)

1) Page 5, Line 8:

For example, the proposed flexible organic flash memory with 40 nm-thick pEGDMA and 16 nm-thick pV3D3 on PET substrates exhibited transistor characteristics that can be switched on at less than 3 V; the device also showed a sufficient voltage shift (ΔV_{th}) of approximately 5 V with V_{prg} and V_{ers} as low as ±10 V (Fig. 2e). Note that these levels of ΔV_{th} and programming/erasing voltages are comparable to those of conventional Si-based flash memory devices.^{2,64} **With $|\Delta n_{fg}| = C_{BDL} \times |\Delta V_{th}| / e$, wherein Δn_{fg} and e refer to the change in carrier density stored in FG and charge of an electron respectively, the observed ΔV_{th} corresponds to Δn_{fg} of 2.1×10¹² cm⁻², which is on par with those of most flash-like memories found in the literature (See Supplementary Table 2).** The reason why Δn_{fg} exhibits little variation among different devices, albeit their

widespread ΔV_{th} , is that a flash memory with lower C_{BDL} generally requires higher operating voltages and thus demands higher ΔV_{th} for proper memory operation, making the $C_{BDL}\Delta V_{th}$ product (and thus Δn_{fg}) remain in the similar range among different systems. In addition, these the flexible memory devices in this study...

4. In Fig 2g, although long data retention has been recorded, the mechanism of data loss should be explained.

(Ans)

What governs the mechanism for loss of data (i.e. stored charges) is the carrier leakage, which is determined ultimately by the J - E characteristics under low electric field. As can be seen in Fig. 1b) – iv), presence of the stored charges in the floating gate (FG) tends to elevate the potential energy of FG and thus results in band bending, but its slope is not large enough to make a triangular barrier; instead, it leads to a trapezoidal barrier. Therefore, the stored charges can be leaked mostly by tunneling from in the low-field regime, the probability of which is very low, even with the assistance of traps (Please refer to Fig. R2 (a) in our response to Reviewer#3's comments for trap-assisted tunneling in the low-field regime.). This kind of tunneling-limited characteristics of TDL can enable long memory retention.

(Revision made in response to Comment#3)

1) Page 5, Line 24:

Moreover, the proposed memory devices exhibited significantly long retention time (t_{ret}), estimated to be 3.2×10^8 sec (≈ 10 years), during which over 50% of the initial ΔV_{th} is projected to be preserved (Fig. 2g). As can be seen in Fig. 1b) – iv), presence of the stored charges in FG tends to yield band bending in energy bands for both TDL and BDL. Because their slopes are still relatively small, however, the field across the insulators remain within the low-field regime. In this case, tunneling probability can be maintained low enough to enable long retention, provided that there are no other leakage paths such as electrical shorts. The observed level of large t_{ret} is thus consistent with tunneling-limited insulating properties of pV3D3 and pEGDMA used in this work. Nonetheless, ... actual retention time in practice. As for programming/erasing speed, the device showed a ΔV_{th} of ca. 2 V at t_{prg} and t_{ers} as low as 10 msec (Fig. 2h), simultaneously fulfilling all the requirements for key performance parameters – reasonably low voltage and short time span for programming and erasing; and sufficiently long retention.

(NOTE) The underlined sentences are those added in this revision in response to another review comment.

5. If possible, it's better to give a cross-sectional image of the edge overlap of CG/BDL/FG/TDL/Channel/S(D) in the memory device as charge leakage may occur through these paths.

(Ans)

Unfortunately, we have no such image due to a relatively large lateral scale associated with a gate electrode patterned by a shadow-masking technique. Nevertheless, there are numerous examples that show conformal coverage of iCVD-based polymers onto a structure with high aspect ratio, as shown in Fig. R4a. This is due to the surface-growing characteristics of iCVD in which polymerization occurs when monomers adsorbed onto a surface meet radicals originating from initiators. Reviewers may be assured that the observed level of memory retention could hardly be expected if there were leakage paths along the edges of the gate electrode.

Another example can be found in the SEM image of pV3D3 coated on a structured Mo/Al₂O₃ layer, which we prepared using the same deposition method for pV3D3 used in this work (Fig. R4b). Note that the tapered shape of the Mo/Al₂O₃ structure mimics the structure near the edge of a bottom electrode. One can clearly confirm that the pV3D3 layer does form a conformal layer over the edge of the structure.

Fig. R4. (a) Example of conformal coverage of an iCVD polymeric film on a high aspect-ratio micro trench. [Image from Baxamusa, Im, and Gleason, Phys. Chem. Chem. Phys., 2009, 11, 5227–5240] (b) SEM image of pV3D3 coated on a structured Mo/Al₂O₃ layer, which mimics the structure near the edge of a bottom electrode.

6. There is a typo in expression 2 in Fig S1, the VFG behind the third equal sign should be revised to VCG.

(Ans)

We thank Reviewer #4 for his or her attention to details. Let us correct the typo.

(Revision made in response to Comment#6)

1) Supplementary Information, Fig. S1, Equation (2):

(original)

$$E_{TDL} = \frac{V_{TDL}}{d_{TDL}} = \frac{V_{FG}}{d_{TDL}} = \frac{\alpha_{CR} V_{FG}}{d_{TDL}} = \frac{k_{BDL} A_{BDL}}{d_{TDL} k_{BDL} A_{BDL} + d_{BDL} k_{TDL} A_{TDL}} V_{CG} \quad (2)$$

→

(revised)

$$E_{TDL} = \frac{V_{TDL}}{d_{TDL}} = \frac{V_{FG}}{d_{TDL}} = \frac{\alpha_{CR} V_{CG}}{d_{TDL}} = \frac{k_{BDL} A_{BDL}}{d_{TDL} k_{BDL} A_{BDL} + d_{BDL} k_{TDL} A_{TDL}} V_{CG} \quad (6)$$

(NOTE) Equation numbers in Fig.S1 have been updated to begin with (5) instead of (1) for succession of equation numbers in Supplementary Information.

7. The 10 years retention time by extrapolation is not acceptable and no one knows what will happen for such a long time with varying temperature and humidity conditions. Therefore it is better to remove such arguments.

(Ans)

In the field of memory researches, using extrapolation of the retention data is a generally accepted method used to predict the long-term retention; however, we agree with Reviewer #4 that it could oversimplify the situation particularly when the materials used are subject to environmental stability issues, etc. Nevertheless, quoting such a value is still important because it provides a good comparison, at least from the electrical perspectives, with the existing technologies. To avoid the concern of overestimation, let us clearly specify that the quoted retention time is the extrapolated value that considers only the electrical aspects and that the actual memory retention is subject to degradation depending on ambient conditions.

(Revision made in response to Comment#6)

1) Abstract

Using their near-ideal dielectric characteristics, we demonstrate flash memories bendable down to a radius of 300 μm that exhibits a **relatively long** projected retention ~~over 10 years~~ with a programming voltage on par with the present industrial standards.

2) Page 5, Line 32

Moreover, the proposed memory devices exhibited significantly long retention time (t_{ret}), estimated to be 3.2×10^8 sec (≈ 10 years), during which over 50% of the initial ΔV_{th} is projected to be preserved (Fig. 2g). As can be seen in Fig. 1b) – iv), presence of the stored charges in FG ... The observed level of large t_{ret} is thus consistent with tunneling-limited insulating properties of pV3D3 and pEGDMA used in this work. **Nonetheless, one should keep in mind that the quoted retention time is the extrapolated value that considers only the electrical aspects and that the actual memory retention is subject to degradation depending on ambient conditions. The long-term test with proper encapsulation should be made to fully assess their actual retention time in practice.** As for programming/erasing speed, the device showed a ΔV_{th} of ca. 2 V at t_{prg} and t_{ers} as low as 10 msec (Fig. 2h), simultaneously fulfilling all the requirements for key performance parameters – reasonably low voltage and short time span for programming and erasing; and sufficiently long retention.

(NOTE) The underlined sentences are those added in this revision in response to another review comment.

3) Page 6, Line 62

In summary, we realized flexible organic flash memory devices on various substrates that exhibit both industry-compatible operating voltage ($\sim 10\text{V}$) and **relatively long projected** retention time (~~~ 10 years~~).

Reviewer #5:

This paper reports on organic flash memory devices based on organic thin-film transistors with a floating gate electrode sandwiched between two dielectric layers.

The properties of these two dielectric layers play an important role in the overall performance of the memory device both in terms of operating voltages and memory retention time.

The main novelty of this work is in the nature and fabrication of the two dielectric layers that are used in these memory devices, namely a tunneling dielectric layer and a charge-blocking layer. In this work the authors employ a deposition technique referred to as initiated chemical vapor deposition (iCVD) that is based on the co-deposition of monomers with an with the initiator to form conformal polymeric layers via free-radical polymerization reactions. This technique developed at MIT in the Gleason group is well-known and has been employed in various organic optoelectronic devices, including organic field-effect transistors (e.g. ref. 51), but appears to be used in the context of memory devices based on OTFTs with a floating gate electrode for the first time.

The amount of data is extensive and the work of excellent quality. The description of the experiments is detailed and the description of the operational principles of the device didactic.

The paper could be further improved if the following comments are addressed:

1. I find the use of the term “platforms” in the title misleading as it is usually used in the contact of an ensemble of technologies that are combined. Here, it looks like the authors meant to say that they demonstrated memory devices on various substrates.
2. Likewise, it is questionable if one should refer to PET and paper substrates as “soft”. Flexible or bendable might be a better term.

(Ans) We thank Reviewer #5 for his or her constructive suggestions. We initially used the term “soft” to include both flexible, elastic, and stretchable materials, but all we did were mainly on flexible materials. We therefore agree with Reviewer#5 that we would rather change it to “flexible.” As for “platform(s)” as well, we agree that “substrate(s)” is a more appropriate term in the present case.

(Revision made in response to Comments#1 and 2)

1) Title

Organic Flash Memory on Various ~~Soft Platforms~~ Flexible Substrates: towards Foldable and Disposable Electronics

2) Abstract

With the scope of electronics being rapidly extended into emerging areas such as wearable or disposable electronics, there grows a demand for a flash memory that is realizable on various ~~soft platforms~~ flexible substrates.

3) In the main text and Supplementary Information

All the “soft” and “platform(s)” used in the main text and SI have been replaced with “flexible” and “substrate(s),” respectively.

3. The authors should give credit to the seminal work on iCVD by proper referencing, as this technique appears to play an important in the presented studies.

(Ans)

As suggested, credit has been given to its inventors (K. K. Gleason and her coworkers), and a review paper by Prof. Gleason has been added.

(Revision made in response to Comment#3)

1) Page 1, Line 52 (Page and line numbers in the *revised* version, unless specified otherwise)

Herein we try to overcome these hurdles and realize highly flexible flash memories by employing thin polymeric insulators grown with initiated chemical vapor deposition (iCVD), a vapor-phase growth technique for polymers pioneered by Gleason and her coworkers.⁵¹

2) Reference added:

51 Baxamusa, S. H., Im, S. G., and Gleason, K. K., Initiated and oxidative chemical vapor deposition: a scalable method for conformal and functional polymer films on real substrates, *Phys. Chem. Chem. Phys.* **11**, 5227–5240 (2009).

4. On line 70, page 1, the authors should replace “molecular structures” with “chemical structures.”

(Ans)

As suggested, the term “molecular structures” has been replaced with “chemical structure.”

(Revision made in response to Comment#4)

1) Page 1, Lines 76

... Two iCVD processed polymer films of poly(1,3,5-trimethyl-1,3,5-trivinyl cyclotrisiloxane) (pV3D3) and poly(ethylene glycol dimethacrylate) (pEGDMA),⁵² whose ~~molecular~~ chemical structures are shown in Fig. 1c,

5. How was the dielectric constant of the TDL determined?

(Ans)

Relative dielectric constants (k) of both TDL and BDL were estimated from capacitance (C) and dielectric thickness (d) measurement using the fact that the capacitance of a capacitor based on two parallel plates is given by:

$$k = \frac{C \times d}{\epsilon_0 \times A} \quad (1)$$

where ϵ_0 and A are the vacuum permittivity and the device area, respectively. Capacitance values were measured with an LCR meter (HP4284A, Agilent) in crossbar metal-insulator-metal (MIM) structures for samples with various thickness ranges. The thickness of dielectric layers was measured for the layers prepared in the same batch with cross-sectional high-resolution transmission electron microscopy (HRTEM). For cross-check the result of thickness measurement, AFM measurement was also done. Detailed information on estimating k of TDL (pV3D3) was not given in the present manuscript because it had been presented in details in authors' previous work. (Methods Section in H. Moon et al. Nat. Mater. 14, 628 (2015), ref. 52)

(Revision made in response to Comment#5)

1) Page 7, Line 57

Electrical characterization of devices. All current density ... Capacitance versus voltage characteristics (C - V) of MIM devices were measured using a precision LCR meter (HP4284A, Agilent). Dielectric constants (k) were estimated by using the relationship among C , k , and thickness of an insulator, following the procedure described in Methods Section in the previous work by Moon et al.⁵² To measure the J - V characteristics of the MIM device....

6. The atmospheric testing conditions should be specified. The electrical properties of C60 are known to degrade in air. Authors should comment on the environmental stability of their devices.

(Ans)

As mentioned by Reviewer #5, some of the materials used in this work are known to be subject to degradation upon exposure to ambient atmosphere. The reason we have used C₆₀ was because C₆₀-based TFTs exhibited a very good run-to-run consistency as well as a reasonably good performance. Relatively low price of C₆₀ and good availability were also helpful in carrying out many batches of experiment. (Please understand some of the high mobility compounds are very expensive, and their availability is often limited.) The run-to-run consistency was very important in our work, as it allowed us to perform a systematic study over many batches of devices fabricated on various kinds of substrates.

Furthermore, one should be reminded that the proposed architecture can be used with little dependence on organic semiconductors. For example, we have recently realized memory devices based on a semiconductor of C8-BTBT – high-performance, air-stable hole-transporting material. The experimental data indicate that the C8-BTBT based non-volatile memory shows high-performance similar to the C₆₀-based memory devices. (See Figure shown below.) Please understand that the work is yet in its early stage and will be published as a separate piece of work.

Nevertheless, we agree with Reviewer #5 that readers should be clearly notified of the environmental stability issues of the materials used in this work. Let us revise the manuscript so that such an issue can be clearly specified.

(Revision made in response to Comment#6)

1) Page 1, Line 71

The proposed non-volatile memory devices are based on C₆₀-based organic TFTs wherein the gate insulator is replaced with a TDL/FG/BDL, as shown in the schematic device structure presented in Fig. 1a. C₆₀ was chosen mainly for its consistency that led to a mobility larger than $1 \text{ cm}^2 \text{ V}^{-1} \text{ s}^{-1}$ with a good run-to-run reproducibility.⁵³⁻⁵⁶ Two iCVD processed polymer films ...

2) References added:

53 Zhang, X.-H., Domercq, B. & Kippelen, B. High-performance and electrically stable C₆₀ organic field-effect transistors. *Applied Physics Letters* **91**, 092114 (2007).

- 54 Anthopoulos, T. D. *et al.* High performance n-channel organic field-effect transistors and ring oscillators based on C₆₀ fullerene films. *Applied Physics Letters* **89**, 213504 (2006).
- 55 Itaka, K. *et al.* High-Mobility C₆₀ Field-Effect Transistors Fabricated on Molecular-Wetting Controlled Substrates. *Advanced Materials* **18**, 1713-1716 (2006).
- 56 Schwabegger, G. *et al.* High mobility, low voltage operating C₆₀ based n-type organic field effect transistors. *Synthetic metals* **161**, 2058-2062 (2011).

3) Figure 2 Caption

Figure 2 | Properties of iCVD processed ultrathin polymer dielectric layers and an organic flash memory made thereof on plastic substrates. **a**, $|J|$ versus E_i characteristics of a Al/pEGDMA(48nm)/Al device. ... **(g)**, and memory speed characteristics for various programming/erasing voltages **(h)**. All the transistor and memory characteristics in Fig. 2 and those appearing later have been measured under controlled atmosphere in an N₂-filled glove box, unless specified otherwise.

4) Page 7 Line 80

In summary, we realized flexible organic flash memory devices on various substrates that exhibit both industry-compatible operating voltage (~10V) and ... The proposed devices on thick PET substrates were found to be able to withstand flexural strain of up to 2.8 %, which is unprecedented for non-volatile memory devices. It is noteworthy that the proposed approach is not specific to C₆₀, which is known to be air-sensitive and thus subject to degradation without encapsulation; some of the air-stable compounds^{80, 81} can also be used provided that their preparation is compatible with flexible substrates of interest. All these advantages are expected to allow nonvolatile memories to be fabricated ...

5) References added:

- 80 Takimiya, K. *et al.* 2, 7-Diphenyl [1] benzothieno [3, 2-b] benzothiophene, a new organic semiconductor for air-stable organic field-effect transistors with mobilities up to 2.0 cm² V⁻¹ s⁻¹. *Journal of the American Chemical Society* **128**, 12604-12605 (2006).
- 81 Zschieschang, U. *et al.* Flexible Low-Voltage Organic Transistors and Circuits Based on a High-Mobility Organic Semiconductor with Good Air Stability. *Advanced Materials* **22**, 982-985 (2010).

6) Page 7 Line 52

Electrical characterization of devices. All current density versus voltage characteristics (J - V) were analyzed using a semiconductor parameter analyzer (HP4155A or HP4156C, Agilent). Electrical characteristics of any devices with C₆₀ were measured under controlled atmosphere in an N₂-filled glove box unless noted otherwise.

7. The strength of this paper is that the overall performance of these devices is impressive as operating voltages similar to those used in Si-based flash memory are demonstrated. The devices yield also long memory retention times. These devices are demonstrated on paper.

A major weakness of this paper is that it fails to provide a clear benchmark with the prior art in organic memory devices with similar geometry. There is substantial of prior art as listed by the authors (refs. 13-36) but a quantitative comparison is lacking. The latter makes the assessment of the novelty and significance of this work difficult. I believe that such a quantitative comparison is needed to justify publication in a journal like Nature Communications.

(Ans)

Comparison of the proposed technology with prior art was shown as a plot in Fig. 3(g) in the original manuscript, but the plot tried to show three kinds of data (voltage, retention time, and flexibility) in a two-dimensional format and had no reference to the source; hence key information can be difficult to recognize and, moreover, room to add some of the technical details of the previous works is rather limited. Let us add a table in Supplementary Information and provide readers with more detailed information on the prior art. We agree with Reviewer #5 that this will provide one with a clear view on the strength and advances of the present work.

(Revision made in response to Comment#6)

1) Page 6, Line 19

Upon comparison with the previous works, the result shown here can be regarded as a significant breakthrough towards realization of highly flexible flash memory that satisfies both low V_{prg} , V_{ers} and long t_{ret} (Fig. 3g Supplementary Table 2 and Supplementary Fig.8.).

2) Supplementary Information

A table comparing the structure and performance of the present organic memory with those of the prior art has been added as Supplementary Table 2. The chart that was originally Fig. 3(g) has been moved to Supplementary Information as Supplementary Fig. 8. Reference to the original sources has been added to the chart for readers to better link what's shown in the table with those in the chart.

Supplementary Table 2.

Summary of the structure and performance of flash-type memory devices based on organic or emerging materials

Lead Author ^[ref]	Substrate	Channel*	BDL (d_{BDL} in nm)/ CSL/ TDL (d_{TDL} in nm) [†]	ΔV_{th} , V_{op} [§] (V, V)	t_{ret} (sec)	Stra-in (%)	Δn_{fg} (10 ¹² cm ⁻²)
Organic flash memory devices on rigid substrates							
Novembre, C. [24]	Si	Pen	SiO ₂ (200)/Au NPs/ -	22, 50	4.5×10 ³	-	2.4
Kim, S.-J. [25]	Si	Pen	SiO ₂ (100)/Au-NPs/ PMMA (40)	34, 80	3×10 ⁴	-	7.3
Kim, Y.-M. [26]	Si	Pen	SiO ₂ (100)/[PE/Au-NPs]/ [HfO ₂ /PVP] (15/54)	14.6, 70	1.6×10 ⁴	-	3.1
Chang, H.-C. [27]	Glass	Pen	HfO/[CuPC NPs/ N-C ₆₀]/ cPVP	4.4, 5	1×10 ⁴	-	1.8
Park, Y. [28]	Si	Pen	SiO ₂ (200)/Graphene/PS (15)	23, 80	~3.2×10 ⁷	-	2.3
Yi, M. [29]	Si	Pen	SiO ₂ (300)/Au-NPs/PMMA (25)	43, 150	9.0×10 ³	-	2.9
Shih, C.-C. [21]	Glass	Pen	cPVP(400)/[ZnO NPs/ PVPK]/-	60,70	1.0×10 ⁴	-	3.4
Shih, C.-C. [30]	Si	Pen	SiO ₂ (100)/[PF & PFBT NPs]/PMAA (30)	35,50	1.0×10 ⁴	-	7.4
Han, S.-T. [33]	Si	Pen	SiO ₂ (100)/[Metal NPs/ MoS ₂ nanosheets]/ Al ₂ O ₃ (5)	27.5, 50	>3.2×10 ⁸	-	5.9
Baeg, K.-J. [19]	Glass	F8T2	PVP (340)/Au-NPs/ PS (22)	30, 80	1.1×10 ⁴	-	3.0
Liu, Z. [35]	Si	P3HT	SiO ₂ (100)/Au-NPs/ PVP (10)	27, 40	2.0×10 ²	-	5.8
Lee, S. [36]	Si	C ₆₀	SiO ₂ (50)/[Ag-NPs/d-SAM]/BCB (12)	4.1, 30	~3.2×10 ⁷	-	1.8
Organic flash memory devices on flexible substrates							
Sekitani, T. [13]	PEN	Pen	[AlO _x /SAM (4/6)]/ Al/ [AlO _x /SAM (4/6)]	2, 6	1.0×10 ⁴	-	8.1
Kaltenbrunner [14]	PEN	Pen	AlO _x (8.5)/Al/ [AlO _x (8.5)/SAM]	2.5, 5	2.0×10 ⁵	-	1.6
Kim, S.-J. [15]	PES	Pen	PVP (400)/Au-NPs/ PVP (20)	15, 90	2.0×10 ⁷	0.1	0.86

Kim S. M. [16]	PEN	Grap.	Al ₂ O ₃ (25) /HfO ₂ / Al ₂ O ₃ (8)	6, 23	4.0×10 ⁵	-	13
Kim, S.-J. [18]	PES	Pen	PVP (400)/Au-NPs/ PVP (10)	10, 90	1.8×10 ⁷	0.1	0.57
Han, S.-T. [22]	PET	Pen	Al ₂ O ₃ (30)/[Au-NPs/ rGO]/ Al ₂ O ₃ (10)	2, 5	>3.2×10 ⁸	0.7	3.6
Lee, S. [This work]	PET	C₆₀	pEDGMA (40)/Al/ pV3D3 (14)	5, 10	>3.2×10⁸	2.8	2.1
Flash memory devices based on emerging channel materials other than organic materials							
Gupta, D. [23]	Si	ZnO	SiO ₂ (100)/Ag-NPs/ -	28.5, 60	1.0×10 ³	-	6.1
Lee, S. Y. [31]	Si	Grap	SiO ₂ (300)/Graphene/ -	120, 150	1.0×10 ⁴	-	8.6
Li, D. [32]	Si	BP	SiO ₂ (300)/MoS ₂ /HBN (25)	60, 20	1.0×10 ³	-	4.3
Bertolazzi S. [34]	Si	MoS ₂	[Al ₂ O ₃ /HfO ₂ (1/30)]/Graphene/ [Al ₂ O ₃ /HfO ₂ (1/6)]	8, 18	2.0×10 ³	-	28

* “Pen” = pentacene; “Grap”= graphene; “BP” = black phosphorous; “F8T2” = Poly(9,9-dioctylfluorene-alt-bithiophene); “P3HT” = Poly(3-hexylthiophene)

† Square brackets were used to indicate the case of multi-component BDL, CSL, or TDL, where CSL refers to a charge storage layer, which could be FG, metal nano-particles (NPs), and so on. Those without values in parentheses indicate that thickness values were not reported in the corresponding papers. “HBN” refers to hexagonal boron nitride.

§ V_{op} = the larger of V_{prog} and V_{erase}

Supplementary Fig. 8.

Comparison of the present work with the TFT-based non-volatile memory devices reported in the literature. The graph classifies the devices according to the kinds of BDL/TDL layers; inorganic or organic ones. Those with ‘ ϵ ’ values correspond to the performance data of flexible memory devices. In those cases, ‘ ϵ ’ refers to the maximum strain tested in the corresponding work. The numbers in square brackets refer to reference numbers in the main text.

Reviewer #2 (Remarks to the Author):

The authors report flexible organic flash memory fabricated on various soft substrates including plastic and paper. Upon optimizing the deposition process and the film thickness of two polymer dielectrics, the resultant devices showed good memory characteristics at low operation voltage of 10 V. Particularly, the memory devices exhibited long projected retention time up to 10 years and at the same time maintained excellent robustness in bending test at a bending radius down to 300 micrometers. The all demonstrated the great potential of these flexible flash memory devices for applications to foldable and disposable electronics. Despite that, however, as the key enablers of these memory devices, the two polymer dielectric layers (TDL and BDL) fabricated by iCVD technique have been published before in Ref. 52. In addition, the device fabrication, characterizations, and analyses reported here are quite conventional with respect to the literature. Therefore, this paper is not original enough to guarantee publication in Nature Communications. Due to the reason, I believe the paper should be submitted to elsewhere.

(Ans)

We thank Reviewer#2 for his or her time spent to carefully evaluate our manuscript and for raising the issue on the originality of the present work. We understand his or her concern, but we do not believe the fact that a certain material was used before should be the reason for judgement against publication even in high-impact journals. If that is the case, many of the papers already in Nature's sister journals or other high-impact journals should have not been published therein. **What is important is to use materials – whether it is known or not - in a right place and/or in a right manner so that one can fully utilize their benefits, unlock their full potential for a given application, and thus generate important advances of significance.**

The work reported in our manuscript constitutes such important advances in flexible or organic electronics fields in that the flexible flash memories reported in this work have all the favorable properties for non-volatile memories: adequate operating voltage, long retention time, and high mechanical durability with near or over ten-fold improvement for all three aspects when compared to prior arts as shown in Figure 3g, which has been moved, in the revised version, to Supplementary Information and accompanied by Supplementary Table 2.

We would like to emphasize that a flash memory is far more complicated to design than thin-film transistors (TFTs). The fact that pV3D3 and pEGDMA were studied for gate dielectrics in TFTs does not guarantee that they can simply be used for memory devices as well. The excellent performance level in the present work was possible only by a careful, rational design that follows the design criteria learned from Si-based devices yet takes into account the features specific to organic devices such as the changes in the effective area for capacitive coupling between programming and erasing (See Fig. 2(c) and (d)). Such a rational design in flexible flash memory devices could not easily be done before due to the scarcity of high quality dielectrics that are readily compatible with flexible electronics. So far, devices with BDL and TDL of thin alumina layers by Han *et al.* showed exhibited a similar level of performance, but the max strain applied was limited to 0.7% due to the limited mechanical properties of alumina layers. That is, the max strain of 2.8% obtained in this work corresponds to enhancement by a factor of four with respect to that of the state of the art. Realizing such a level of flexibility in flash memories – with almost no compromise in performance – should not be taken for granted and in fact should be regarded as a significant achievement.

It may sound ironic, but the fact that (flexible) flash-like organic memory devices were realized based on a *rational design approach* by itself may be regarded as novel and significant. Looking back the history of organic electronics, one can note that most of the significant milestones in organic electronics were made when a rational design approaches were used that took lessons from their inorganic counterparts yet took into account the properties specific to organic devices. The present work may be regarded as another example of such cases.

Response to Reviewers' Comments

1. Response to Reviewers' comments

As shown below, all the reviewers recommended the present manuscript to be accepted 'as is' and thus did not indicate any further content revisions to make; therefore, we have focused on revising the manuscript according to the formatting guidelines kindly provided by Editor.

Reviewer #3 (Remarks to the Author):

I've originally already recommended publication of the manuscript by Seunghyup Yoo and co-workers with minor revisions. The manuscript discloses new impressive benchmarks reached by rational materials selection and device design for floating-gate memory applications. This will likely stimulate new waves of research in different directions. In the revised manuscript now, the authors have further refined their manuscript and clarified several technical issues. **I have no other queries, and can therefore recommend publication in its present form.**

Reviewer #4 (Remarks to the Author):

The authors revised the manuscript as per reviewers' recommendation therefore **I recommend for publication.**

Reviewer #5 (Remarks to the Author):

In their revised version, the authors have thoroughly addressed all of my comments and recommendations. The quality of the paper has improved significantly. **It meets now, in my opinion, the high standards for publication in Nature Communications.**